# GDPval: Evaluating AI Model Performance on Real-World Economically Valuable Tasks

**Tejal Patwardhan**[*]   **Rachel Dias**[*]   **Elizabeth Proehl**[*]   **Grace Kim**[*]   **Michele Wang**[*]

**Olivia Watkins**[*]   **Simón Posada Fishman**[*]   **Marwan Aljubeh**[*]   **Phoebe Thacker**[*]

Laurance Fauconnet    Natalie S. Kim    Patrick Chao    Samuel Miserendino

Gildas Chabot    David Li    Michael Sharman    Alexandra Barr    Amelia Glaese    Jerry Tworek

OpenAI

## Abstract

We introduce GDPval, a benchmark evaluating AI model capabilities on real-world economically valuable digital knowledge-work tasks. GDPval covers the majority of US Department of Labor O*NET Work Activities for 44 occupations across the top 9 sectors contributing to U.S. GDP (Gross Domestic Product). Tasks are constructed from the representative work of industry professionals with an average of 14 years of experience. We find that frontier model performance on GDPval is improving roughly linearly over time, and that the current best frontier models are approaching industry experts in deliverable quality. We analyze the potential for frontier models, when paired with human oversight, to perform GDPval tasks cheaper and faster than unaided experts. We also demonstrate that increased reasoning effort, increased task context, and increased scaffolding improves model performance on GDPval. Finally, we open-source a gold subset of 220 tasks and provide a public automated grading service to facilitate future research in understanding real-world model capabilities.

## 1 Introduction

There is growing debate about how increasingly capable AI models could affect the labor market—whether by automating specific tasks, replacing entire occupations, or creating entirely new kinds of work (Brynjolfsson et al., 2025; Chen et al., 2025). Current approaches to measure the economic impact of AI focus on indicators such as adoption rates, usage patterns, and GDP growth attributed to AI (Chatterji et al., 2025; Tamkin et al., 2024; Appel et al., 2025; Acemoglu, 2025; Bick et al., 2024). However, historical evidence from technological shifts—such as electricity, airplanes, and computers—shows that the transition from invention to economy-wide permeation often takes years or even decades, requiring regulatory, cultural, and procedural changes (David, 1990; Brynjolfsson & Hitt, 2000; Brynjolfsson et al., 2017; Dwivedi et al., 2021; Solow, 1987). Therefore, while informative when available, these methods are lagging indicators of AI impacts. We consider an alternate method for understanding the potential economic impacts of AI: directly measuring AI model capabilities. AI capability evaluations can provide clearer, more directly attributable evidence about model abilities, allowing us to assess economic relevance ahead of widespread adoption.

Our paper introduces the first version of GDPval, a benchmark evaluating AI model performance on real-world economically valuable tasks. GDPval covers the top 9 sectors contributing to U.S. GDP (Gross Domestic Product), with at least 30 tasks per occupation in the full set (and 5 tasks per occupation in the gold subset), across 44 occupations. Each task is constructed based on actual work product created by an expert professional. Given the complexity of automatically grading these tasks, our primary evaluation metric is head-to-head human expert comparison. We also provide an experimental automated grader service for the 220 open-sourced gold subset of tasks. Future GDPval iterations will incorporate greater breadth, realism, interactivity, and contextual nuance.

The initial version of GDPval offers several advantages over existing AI model evaluations:

---

[*]Equal contribution.

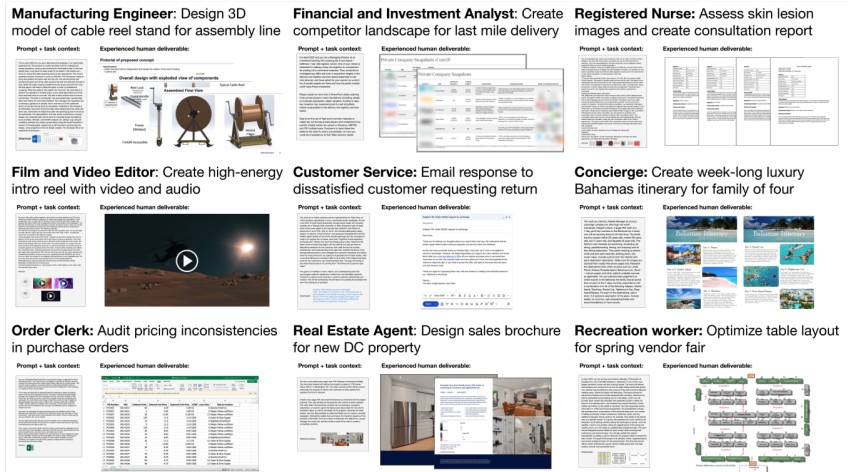

Figure 1: Example GDPval tasks from full set

- **Realism**: Unlike AI benchmarks in the style of an academic test that focus on reasoning difficulty (e.g., Phan et al. (2025); Hendrycks et al. (2020); Rein et al. (2023); Liu et al. (2023)), tasks are based on actual work product from industry experts, validated through multiple rounds of review, and tied to time and cost required for completion.
- **Representative breadth:** Unlike AI evaluations focused on specific domains like software engineering (e.g., Miserendino et al. (2025)), the GDPval full set covers 1,320 tasks across 44 occupations, sourced to cover the majority of Work Activities tracked by O*NET for each occupation U.S. Department of Labor, Employment and Training Administration (2024). This top-down approach allows for representativeness of tasks across occupations. We also build on production AI usage analyses (e.g., Tamkin et al. (2024); Chatterji et al. (2025); Appel et al. (2025)) to cover areas where model adoption is still emerging.
- **Computer use and multi-modality**: Tasks require manipulating a variety of formats (e.g., CAD design files, photos, video, audio, social media posts, diagrams, slide decks, spreadsheets, and customer support conversations). Each task also requires parsing through up to 17 reference files in the gold subset, and 38 in the full set.
- **Subjectivity**: In addition to correctness, expert graders often consider subjective factors such as structure, style, format, aesthetics, and relevance. Our dataset also therefore serves as a helpful testbed to assess automated grader performance.
- **No "upper limit"**: Unlike metrics that could saturate quickly, our primary metric is win rate, which allows for continuous evaluation. Currently, we compare model outputs against a human expert baseline, but we could replace our baseline with increasingly strong models over time and keep evaluating.
- **Long-horizon difficulty**: Tasks require an average of 7 hours of work for an expert professional to complete. On the high end, tasks span up to multiple weeks of work.

## 2 TASK CREATION

We first identify the sectors that contribute most to U.S. GDP, then source tasks drawn from the highest-earning knowledge work occupations within those sectors.

### 2.1 PRIORITIZING OCCUPATIONS

GDPval covers tasks from 9 sectors and 44 occupations that collectively earn $3T annually. We detail below the methodology behind our initial version.

To choose the initial occupations, we:

1. **Selected sectors that contribute over 5% to US GDP** as determined by Q2 2024 Value Added by Industry as a Percentage of Gross Domestic Product (see Federal Reserve Bank

of St. Louis (2025)). These 9 sectors are shown in Table 1. Because our current data budget did not allow for collection of tasks across all industries, we start with the highest-contribution industries to GDP in this the first iteration of GDPval, but we plan to expand to more industries in future iterations.

2. **Selected the 5 occupations[1] within each sector that contribute most to total wages and compensation and are predominantly digital.** Because we could not collect tasks for every occupation in each industry due to budget constraints, we started by prioritizing the top 5 in each industry. We took a task-based approach to determining if an occupation should be classified as "predominantly digital." Specifically, we identified all tasks for an occupation from O*NET, a database of occupational data, definitions and tasks from the U.S. Department of Labor. Similar to Eloundou et al. (2023), we prompted GPT-4o to classify each task as digital or non-digital, and then classified the overall occupation as digital if at least 60% of its component tasks were digital. To calculate this percentage, we weighted tasks by the "relevance," "importance," and "frequency" scores for each task reported in O*NET Task Ratings.

We further validated the representativeness of our digital tasks measure by benchmarking it against the Acemoglu & Autor (2011) task content framework. The correlations we observe—digital tasks increasing with non-routine cognitive content and decreasing with routine and manual content—demonstrate alignment with established economic measures of work, as per appendix A.8.6.

For wage and occupation data, we used the Bureau of Labor Statistics May 2024 national employment and wage estimates to calculate total wages for 831 occupations (U.S. Bureau of Labor Statistics (2025)) and further detailed in appendix A.8.

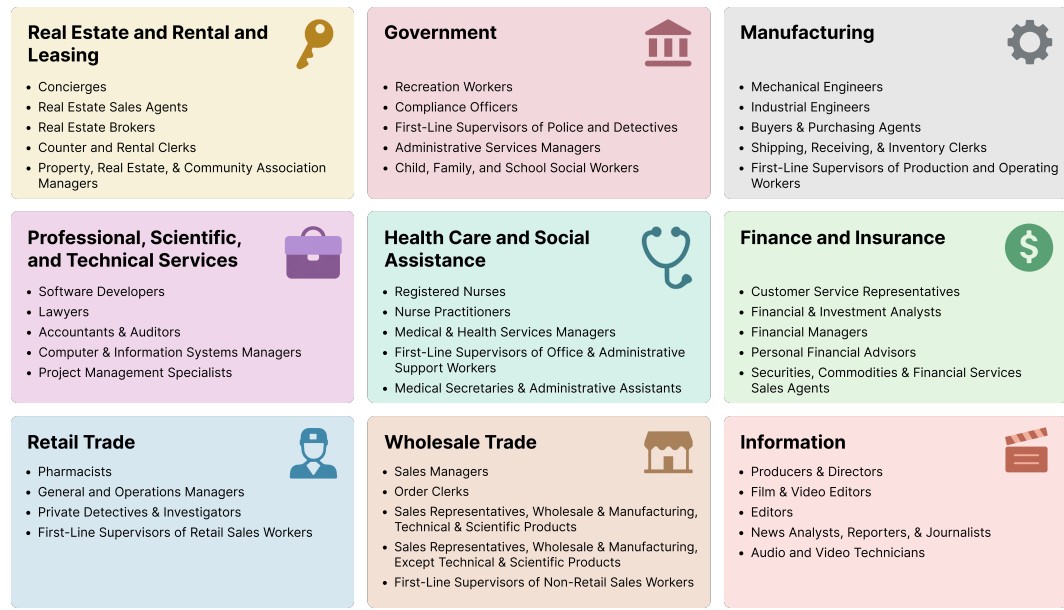

Figure 2: GDPval includes real-world work from 44 occupations.

## 2.2 EXPERT RECRUITMENT

We recruited expert industry professionals to create realistic tasks based on their professional work experience. Experts were required to have a minimum of 4 years of professional experience in their occupation and a strong resume with a demonstrated history of professional recognition, promotion, and management responsibilities. The average expert had 14 years of experience. We further

---

[1] We assigned occupations to sectors by using the 2023 BLS National Employment Matrix from of Labor Statistics (2025) to map occupations to sectors by identifying the sector with the highest employment for each occupation. For more detail, see appendix A.8.

required experts to pass a video interview, a background check, a training and a quiz to participate in the project. Experts were well compensated for their time and experience. Some of the prior employers of our industry experts include: Accenture, Aetna, Apple, AXA Advisors, Bank of America, Barclays, BBC News, Boeing, Budget Rent a Car, Capital One, Centers for Disease Control and Prevention, Citigroup, Condé Nast, CVS Pharmacy, U.S. Department of Defense, Disney, Douglas Elliman, E*TRADE, Federal Trade Commission, General Electric, Goldman Sachs, Google, Guggenheim Partners, HBO, IBM, JPMorgan Chase, Johnson & Johnson, Kmart, Kirkland & Ellis LLP, LinkedIn, Lockheed Martin, Macy's, Massachusetts General Hospital, Meta, Microsoft, Morgan Stanley, National Park Service, NFL Network, Oracle, Paul, Weiss, Rifkind, Wharton & Garrison LLP, Prudential, PwC, Raytheon, Sally Beauty, Samsung, SAP, Scientific American, Sotheby's, Telegraph Media Group, Thermo Fisher Scientific, TIME, Twilio, U.S. Department of Justice, United States Air Force, United States Postal Service, Walgreens, Wells Fargo, White & Case LLP, and Whole Foods. More information can be found in appendix A.6.

## 2.3 TASK CREATION

Each GDPval task consists of two primary components: a request (often with reference files) and a deliverable (work product). Tasks are based on work performed by experts in their professional experience and then mapped to O*NET occupational tasks for their occupation to ensure broad and representative coverage across tasks (National Center for O*NET Development, 2025). Experts created multiple tasks, and multiple experts created tasks for each occupation. More details on task characteristics can be found in appendix A.4. To assess task quality, we also asked occupational experts to rate each task on its difficulty, representativeness, time to complete, and overall quality against real-world standards for their occupation. Each task's dollar value was estimated by multiplying the estimated completion time (self-reported by the expert who wrote the task) by median hourly wages for the corresponding occupation from OEWS data (U.S. Bureau of Labor Statistics, 2025).

## 2.4 TASK QUALITY CONTROL PIPELINE

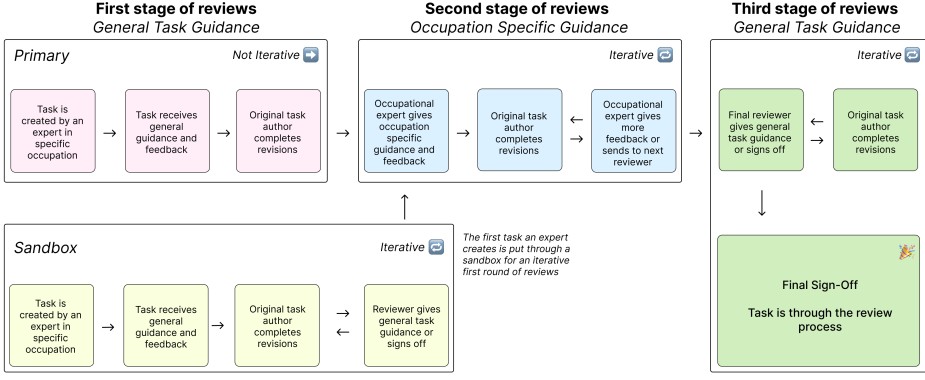

Figure 3: Tasks undergo multiple rounds of review to ensure realism and quality. More detail on the review pipeline in appendix A.5

All 1,320 tasks in the full GDPval set went through an iterative review pipeline involving both automated model-based screening and multiple stages of human expert review. Each task received an average of five human reviews (with a minimum of three reviews). Across all stages of review, experts provided detailed comments, and tasks were iteratively revised before subsequent reviews to enhance quality and representativeness, as detailed in appendix A.5.

## 2.5 HUMAN EXPERT GRADING AND AUTOMATED GRADING

To grade the 220 open-sourced gold subset, we conducted blinded expert pairwise comparisons, where experts in the relevant occupation were presented with unlabeled work deliverables, including

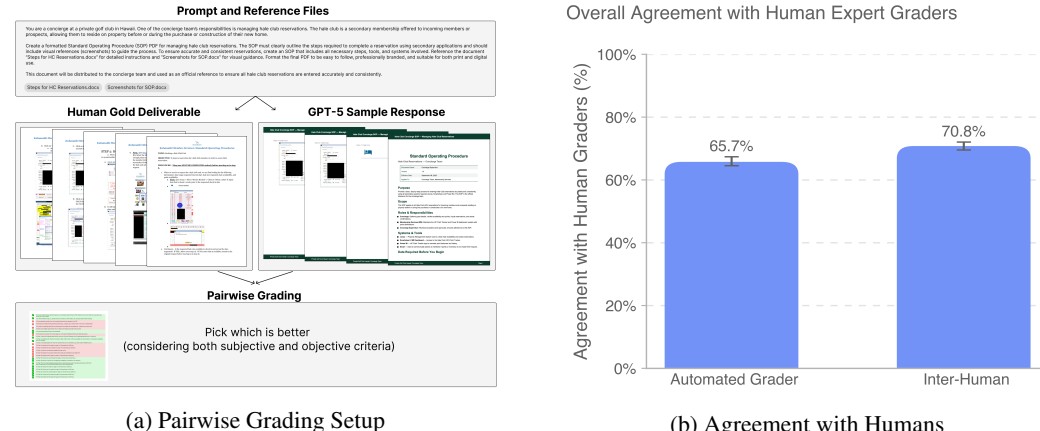

(a) Pairwise Grading Setup  (b) Agreement with Humans

Figure 4: GDPval uses pairwise expert comparisons for grading. We also create an experimental automated grader. We find that automated grader agreement is within 5% of human inter-rater agreement on the GDPval gold subset.

those from models and from human experts, and asked to rank them based on which better fulfilled the task request given its reference files.

On average, grading each comparison for the gold subset took over an hour. Additional occupational experts were sourced to grade human and model deliverables. Experts provided detailed justifications for their choices and rankings, which enabled us to compute our headline win rates for various models compared to the human expert completion.

For the gold subset, we trained an experimental grading model to perform pairwise comparisons in the style of industry professional experts. The automated grader is presented with the request and reference files, two deliverables, and a rubric, and asked to pick which completion is better. Although limited, the automated grader is faster and cheaper than expert grading, and achieves 66% agreement with human expert graders, only 5% below human expert inter-rating agreement of 71%. Further detail is in appendix A.7.

## 3 EXPERIMENTS AND RESULTS

### 3.1 HEADLINE RESULTS

We evaluated GPT-4o, o4-mini, o3, GPT-5, Claude Opus 4.1, Gemini 2.5 Pro, and Grok 4 using blind pairwise comparisons by professional industry experts.[2]. We sampled Claude, Gemini and Grok via the UI to enable the maximum GDPval-relevant performance. For example, for Claude, we wanted to evaluate its upgraded file creation and analysis feature (Anthropic, 2025). For the OpenAI models, we enabled the web search tool and the code interpreter tool, with background sampling. We also preinstalled several libraries not available in the base image, see appendix A.7.5. For plots shown, we sampled each model 3 times for each prompt, and then had 3 different human graders grade each sample (yielding 9 comparisons per prompt, per model, across 220 tasks). Claude Opus 4.1 was the best performing model on the GDPval gold subset, excelling in particular on aesthetics (e.g., document formatting, slide layout), while GPT-5 excelled in particular on instruction following (e.g., producing the requested deliverable format, including all requested sections) as per fig. 8. This distinction is also shown in appendix A.2.4, where GPT-5 performs better on pure text, while Claude performs better on file types like .pdf, .xslx, and .ppt, demonstrating better visual and aesthetic abilities. We caveat also that the occupations and task types covered by text tend to be different than

---

[2]We aimed to keep comparisons as blind as possible, but model samples may still have been identifiable due to stylistic differences. OpenAI outputs often used em dashes, Claude outputs frequently adopted first-person phrasing, and Grok occasionally referred to itself as Grok. Although filenames were scrubbed of model identifiers, to preserve sample identity, we did not alter style or content, so experts may still have been able to infer model origins

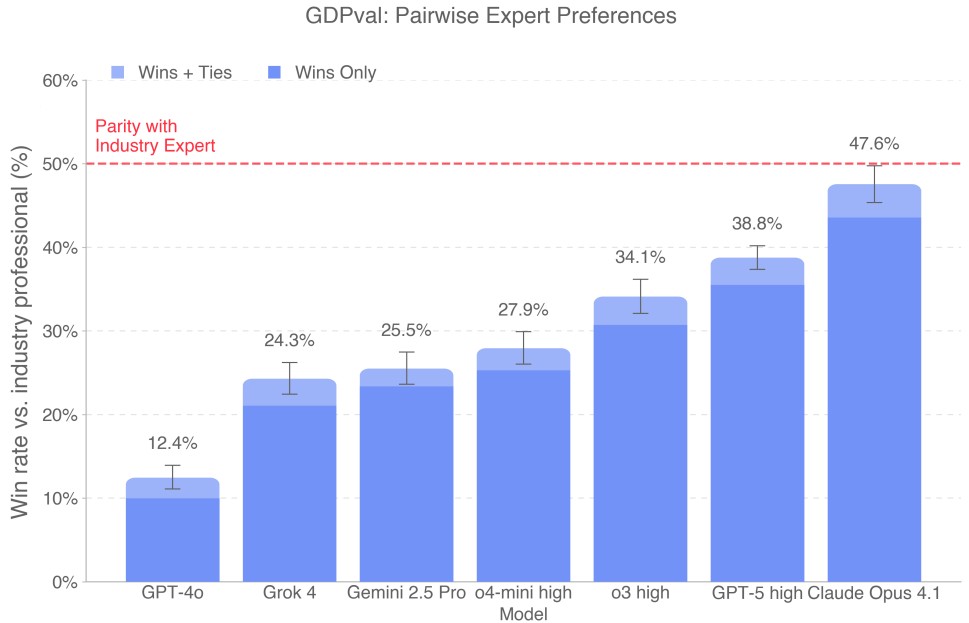

Figure 5: On human pairwise comparisons, models are beginning to approach parity with industry experts on the GDPval gold subset.

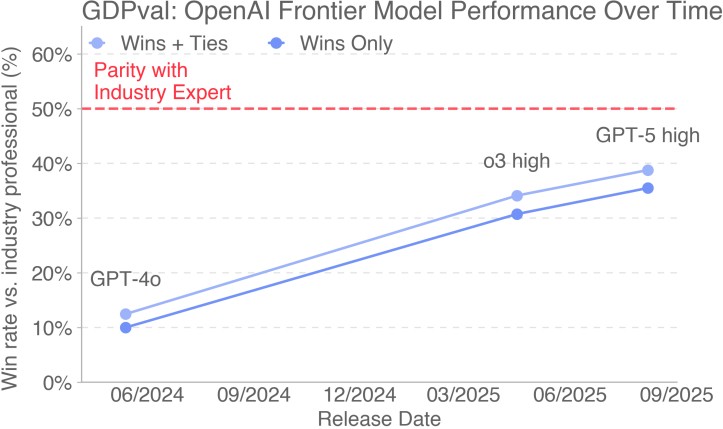

Figure 6: Performance of OpenAI frontier models increased roughly linearly over time on the GDP-val gold subset.

those that involve multi-modal. See table 3 for more information about sampling files. In fig. 5, on the GDPval gold subset, 47.6% of deliverables by Claude Opus 4.1 were graded as better than (wins) or as good as (ties) the human deliverable. Model deliverables outperformed or matched expert humans' deliverables in just under half the tasks.

## 3.2 SPEED AND COST COMPARISON

We analyzed several scenarios to understand the potential speed and cost savings ratio of frontier models on the GDPval gold subset tasks in appendix A.2.1[3]. In the scenarios analyzed, incorporating frontier AI models into expert workflows showed the potential to save time and money relative to

---

[3]We were not able to obtain cost estimates for Claude, Gemini, and Grok. Claude's new files UI handles files more reliably than the API but does not expose time or token metadata, preventing comparable cost/time

unaided experts. Fig 7 summarizes expected savings under a "try using the model and if still unsatisfactory, fix it yourself" setup. Here, an expert human samples from a model, reviews outputs, and if unsatisfactory, resamples and repeats. If no satisfactory output is obtained, the human completes the task themselves. Under this setup, as well as other setups (e.g., directly using model outputs, trying the model just once before doing work directly), model assistance can potentially save the expert time and money. We report results using GPT-family models as a stand-in for frontier models generally. Our intent in these experiments was to study human versus model versus human with model, not to compare cost and time across model families.

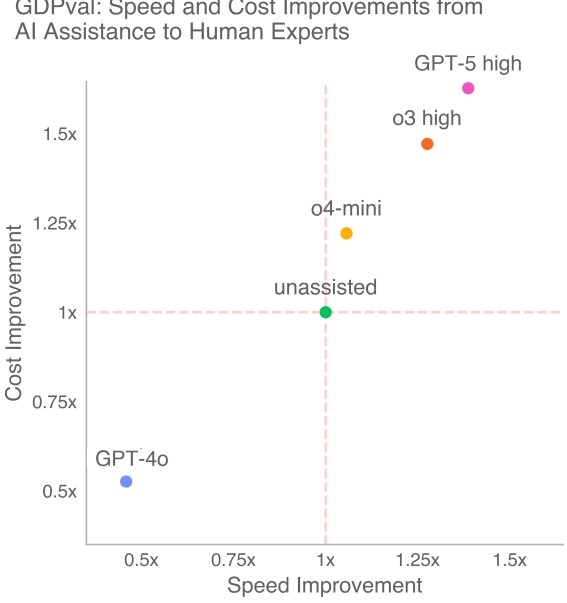

Figure 7: In the scenarios we analyze, models show the potential to save time and money by coupling AI assistance with expert human oversight. Here, we show speed and cost savings from a "try $n$ times, and if still unsatisfactory, fix it yourself" approach as detailed in appendix A.2.1.

## 3.3 MODEL STRENGTHS AND WEAKNESSES

We built a clustering pipeline to analyze why experts preferred or rejected GPT-5 high, Claude Opus 4.1, Gemini 2.5 Pro, and Grok 4 deliverables as shown in fig. 8. Samples were clustered using the natural language rationales expert human graders wrote for their choices. Grok and Gemini most often lost due to instruction-following failures and frequently promised but failed to provide deliverables, ignored reference data, or used the wrong format. Claude and GPT-5 high showed better instruction following, though all models sometimes hallucinated data or miscalculated. A more detailed table of failure modes is available in A.2.6.

---

analysis. Gemini and Grok had similar API file-handling limitations so we sampled completions via their first party platforms in order to solicit the best possible performance.

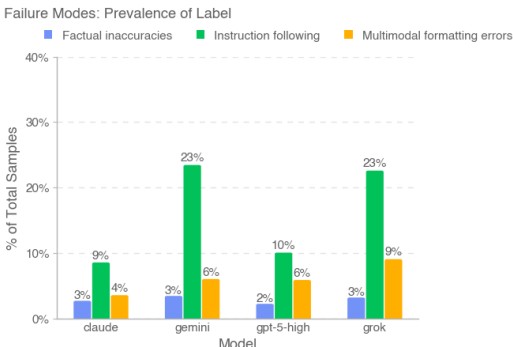

Figure 8: Across models, experts most often preferred the human deliverable because models failed to fully follow instructions on GDPval tasks.

### 3.4 INCREASING REASONING EFFORT AND SCAFFOLDING

To understand the impact of reasoning effort on model performance, we ran GDPval on the o3 and GPT-5 models at low, medium, and high reasoning effort. We found that additional reasoning effort improved performance by up 4.3 percentage points for OpenAI o3 and 6.1 percentage points for GPT-5.

We were also interested in measuring how easily we could improve model capabilities with prompts. For example, many of the observed GPT-5 failure modes were due to obvious formatting errors. We created a prompt which encouraged GPT-5 to rigorously check deliverables for correctness, check layouts by rendering files as images, avoid nonstandard unicode characters, and avoid excess verbosity. The prompt applies generally to multimodal economic tasks and is not overfit to any given question (see appendix A.3 for details). We also improved agent scaffolding by enabling GET requests in the container and performing best-of-N sampling with N=4 and a GPT-5 judge[4].

Prompting fully eliminated black-square artifacts from GPT-5 responses, which previously affected over half of generated PDFs, and reduced egregious formatting errors in PowerPoint files from 86% to 64%. This can be partially attributed to a sharp increase in agents using their multi-modal capabilities to inspect deliverables ($15\% \rightarrow 97\%$). Prompting also improved human preference win rates by 5 percentage points in Figure 9b. These easy performance gains suggest there are paths to agent improvement on GDPval tasks by training or scaffolding them to be more thorough and take full advantage of their multimodal capabilities.

## 4 OPEN-SOURCING

We open-source the prompts and reference files in our 220-task gold subset on Hugging Face (dataset: GDPval). While human expert comparison is still our recommended method of grading, we make an experimental automated grader publicly available as well, which is linked from the Hugging Fave, and also available at evals.openai.com. Please note that the tasks in the open sourced set have been scrubbed of information that could be used to identify the expert who wrote the task. We also note that, as a result of limitations with our automated grader, we don't provide automated grading results for all tasks in the gold subset. Further disclaimers about the open source gold subset are in appendix A.1.3.

## 5 LIMITATIONS

**Focus on self-contained knowledge work:** Tasks in the initial version of GDPval are oriented around knowledge work that can be performed on a computer, particularly around digital deliverables. Manual labor and physical tasks are not included in the current version. Moreover, tasks that

---

[4]Empirical evidence shows that models often favor their own responses. See A.7 for further discussion of limitations with the autograder.

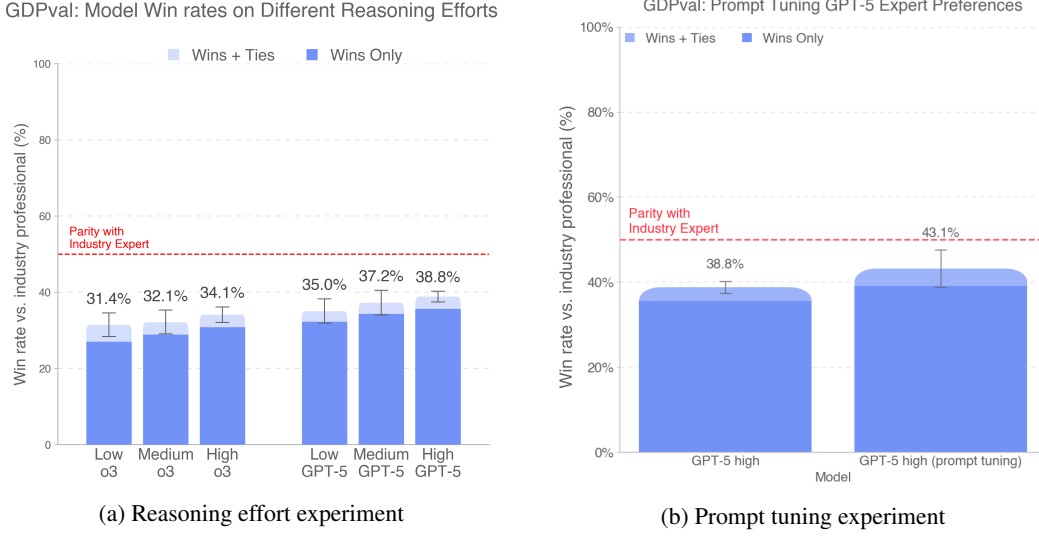

(a) Reasoning effort experiment

(b) Prompt tuning experiment

Figure 9: Model performance improves predictably with increasing reasoning effort. Prompt-tuning and scaffolding improvements also increase GPT-5 performance.

involve extensive tacit knowledge, access to personally identifiable information, use of proprietary software tools, or communication between individuals are out of scope for the current evaluation. We aim to build on this in future versions of the evaluation.

**Tasks are precisely-specified and one-shot, not interactive:** For GDPval, we provide the full context of the task in the prompt, but in real life it often takes effort to figure out the full context of a task and understand what to work on. We are working on improvements to GDPval that involve more interactivity and contextual realism. In the meantime, the experiment in the "Under-contextualized GDPval" section (appendix A.2.7) demonstrates how model performance degrades with less context.

**Dataset size:** The GDPval full set currently consists of only 44 occupations and 30 total tasks per occupation. It is therefore a limited, initial cut of knowledge work tasks, not a comprehensive evaluation of all possible occupational tasks. We are expanding the dataset size.

**Grader performance:** Our current automated grader has a number of limitations compared to human expert graders. More details about the automated grader are available in the appendix A.7.2.

**Cost:** Constructing and running our evaluation is expensive, particularly with industry expert graders. For this reason, we make an automated grader proxy available, but do not consider it a full substitute for industry expert graders.

## 6 CONCLUSION

In GDPval, we contribute the following:

1. **Dataset**: We create a new evaluation dataset (GDPval) measuring real-world, economically valuable knowledge-work tasks.

2. **Capability benchmarking:** We analyze quality, speed and cost of deliverables across human industry experts and frontier AI models.

3. **Experiments:** We test how results shift with differing reasoning effort, prompting, scaffolding, and context.

4. **Open-sourcing:** We open-source 220 tasks as part of our gold subset which includes prompts and reference files.

5. **Automated grader:** We release an experimental automated grader to improve accessibility of grading.

We hope this work contributes to the science of tracking model progress, so that we have better data to assess the social impacts of AI models.

## 7 ETHICS STATEMENT

**Human participants.** This work relies on tasks and judgments from industry experts (experts, reviewers, and graders). All participants were told the purpose of the project, the types of data being collected, and how the data would be used for research. We don't believe that our study meets the definition of human-subjects research because we were studying models, not human subjects. Experts served as professional contractors producing and evaluating work outputs. This is analogous to standard data-labeling pipelines. Participation was voluntary, with the option to withdraw at any time. Experts were compensated for their time and expertise; compensation was not contingent on model outcomes. More detail is available in A.6.

**Privacy, confidentiality, and data governance.** We instructed contributors not to submit personally identifiable information (PII), material non-public information (MNPI), and confidential or proprietary content in any submissions. Reviewers flagged overlooked information, which was subsequently replaced with realistic, fictional data by contributors. We do not release raw deliverables that could reveal sensitive information. O*NET, OEWS and BLS data were used in accordance with their terms.

**Potential harms and misuse.** GDPval measures model capabilities on economically valuable knowledge work tasks. Results could be misused to make premature claims about job replacement or to target specific roles for automation without context. To mitigate this, we emphasize limitations (capability versus adoption; subjectivity; file and tool constraints) and report results with confidence intervals. These results are research findings and are not intended to inform or justify personnel decisions (e.g., hiring, firing, promotion, or compensation). The benchmark excludes dangerous or clearly harmful task types.

## ACKNOWLEDGEMENTS

We thank Addea Gupta, AJ Ostrow, Aleksander Madry, Ally Bennett, Alexander Wei, Becky Waite, Ben Gaffney, Brad Lightcap, Casey Chu, Cassandra Duchan Solis, Charlotte Cole, Dane Stuckey, Eric Wallace, Erik Ritter, Evan Mays, Fidji Simo, Gideon Myles, Hannah Wong, Isa Fulford, Jakub Pachocki, James Lennon, Jared Pochtar, Jason Kwon, Jordan Frand, Justin Wang, Julia Steele, Karthik Rangarajan, Kevin Liu, Larry Summers, Leyton Ho, Leo Liu, Leon Maksin, Livvy Pierce, Lindsay McCallum, Manoli Liodakis, Mark Chen, Max Schwarzer, Miles Palley, Miles Wang, Nakul Khanna, Nat McAleese, Nicholas Carlini, Nick Otis, Nick Ryder, Noel Bundick, Paul Radulovic, Phillip Guo, Prashanth R, Rachel Brown, Raoul de Liedekerke, Robert Rotsted, Ronnie Chatterji, Ryan Kaufman, Sam Altman, Sam Bowman, Sherwin Wu, Tom Cunningham, Tom Stasi, Tony Song, Trevor Creech, Wenda Zhou, Wenlei Xie, Wyatt Thompson, and Yara Khakbaz for discussion, assistance, and review. We'd especially like to thank the industry experts who contributed their time and expertise to GDPval, without whom this work would not have been possible.

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

## A    APPENDIX

### A.1    DISCLOSURES

#### A.1.1    AI DISCLOSURE

We used AI models to help with our literature review and with tweaking language in the paper. We also used AI coding assistants as part of our regular engineering workflows (e.g., to help find and fix bugs).

#### A.1.2    SENSITIVE CONTENT AND POLITICAL CONTENT DISCLOSURE

Some tasks in GDPval include NSFW content, including themes such as sex, alcohol, vulgar language, and political content. We chose to keep these tasks as they reflect real themes addressed in various occupations (e.g., film, literature, law, politics). We do not endorse the particular actions or views in any of the content.

#### A.1.3    THIRD-PARTY REFERENCES DISCLOSURE

GDPval contains limited references to third-party brands and trademarks solely for research and evaluation purposes. No affiliation or endorsement is intended or implied. All trademarks are the property of their respective owners. Some images and videos in this dataset feature AI-generated individuals and real people who have provided permission. Names and identifying references to private individuals in GDPval are fictitious. Any resemblance to actual persons or entities is purely coincidental.

## A.2 Additional Detail on Experimental Results

### A.2.1 Speed and Cost Analysis, continued

We use the following definitions:

1. Human expert professional completion time $H_T$ is the time taken by a human expert professional to complete a task, based on validated self-reported time to complete[5]. To calculate human expert professional completion cost $H_C$, we multiplied the reported task completion hours per occupation by the median hourly wage for each occupation from the U.S. Bureau of Labor Statistics (2025)[6]. On average, on our 220 gold subset $H_T = 404$ minutes and $H_C = \$361$.

2. Human expert professional review time $R_T$ is an estimate of the time taken to assess a model deliverable by a human expert grader. We observe this from our task monitoring software, averaging the time taken to grade for the first time each human expert was asked to grade that question. On average, $R_T = 109$ minutes, and associated human expert professional review cost $R_C$ is on average $86, where $R_C$ is again calculated based on time taken multiplied by median wage data.

3. Model completion time $M_T$ is the time taken for the model to complete a deliverable and $M_C$ is the associated completion cost, based on empirical API speed and cost for the model to complete the deliverable when given a prompt [7].

4. Model win rate $w$ is how often the model deliverable is rated better than the human deliverable by the human expert grader.

We then calculate the following ratios:

1. **Naive ratio:** To measure the ratio of human deliverable versus model deliverable, without accounting for any quality differences or implementation times, we simply divide the average task completion time for a human by the average sampling time for a model: $H_T/M_T$, and analogously for cost: $H_C/M_C$.

2. **Try 1 time, then fix it ratio:** To calculate the time with this method, we take the sampling time for the model, add review time $R_T$ for an expert to assess quality, and then with probability $(1 - w_i)$ add in the human completion time for any fixes needed for that model for a task $i$, to obtain $T_{1,i}$ and analogously $C_{1,i}$:

$$\mathbb{E}[T_{1,i}] = M_{T,i} + R_{T,i} + (1 - w_i)H_{T,i} \tag{1}$$
$$\mathbb{E}[C_{1,i}] = M_{C,i} + R_{C,i} + (1 - w_i)H_{C,i} \tag{2}$$

   The average time spent is $T_1 = \mathbb{E}[T_{1,i}]$, marginalizing over all tasks $i$, similarly with $C_1$. This proxies the setup where a human tries using GPT-5 for a task, assesses its quality, and then does the task themselves if the deliverable quality is below their quality bar. Our plug-in estimate of the time savings ratio is: $H_T/(M_T + R_T + (1 - w)H_T) = H_T/\hat{T}_1$, where we use the empirical mean $\hat{T}_1$. The analogous cost ratio is $H_C/(M_C + R_C + (1 - w)H_C)$.

3. **Try $n$ times, then fix it ratio:** To calculate the time with this method, we take the sampling time for the model, add review time $R_T$ for an expert to assess quality, and then add in the human completion time for any fixes needed for that model (based on $1 - w_i$) [8]. We repeat this across $n$ resamples and re-assess steps before the human steps in to fix it:

---

[5]During submission, experts self-reported the real-world time required to complete each task. Multiple occupational reviewers independently validated these times, correcting errors. Because times were self-reported, it is possible that experts under-estimated or over-estimated time taken

[6]Because our experts were recruited specifically for being highly experienced in their field, these wage estimates likely underestimate their true market cost.

[7]For each task, we collected three API completions per model and averaged the observed response times recorded in the API metadata. We also recorded the average invoiced cost per task.

[8]We are over-penalizing the model here, because the win rate after each completion likely goes up (because the professional will adjust the prompt to the model to fix the errors) and the review time also goes down as the professional gets more comfortable with the task.

$$\mathbb{E}[T_{n,i}] = \sum_{k=1}^{n} \left( (1-w_i)^{k-1}(M_{T,i} + R_{T,i}) \right) + (1-w_i)^n H_{T,i} \tag{3}$$

$$= (M_{T,i} + R_{T,i}) \frac{1-(1-w_i)^n}{w_i} + (1-w_i)^n H_{T,i} \tag{4}$$

$$\mathbb{E}[C_{n,i}] = \sum_{k=1}^{n} \left( (1-w_i)^{k-1}(M_{C,i} + R_{C,i}) \right) + (1-w_i)^n H_{C,i} \tag{5}$$

$$= (M_{C,i} + R_{C,i}) \frac{1-(1-w_i)^n}{w_i} + (1-w_i)^n H_{C,i} \tag{6}$$

This proxies the setup where a human tries $n$ rounds of using GPT-5 for a task, then assesses its quality each time, and then does the task themselves if the model quality is below their quality bar after all attempts. As before, the average time spent is $T_n = \mathbb{E}[T_{n,i}]$, marginalizing over all tasks $i$, similarly with $C_n$. Therefore, as $n \to \infty$, with $w > 0$, the time savings are $H_T/((M_T + R_T)/w)$ times faster and cost savings are $H_C/((M_C + R_C)/w)$ times cheaper than human experts.

Table 1: Speed and cost improvements under different review strategies.

| Model | Win rate | Speed improvement | | | Cost improvement | | |
|---|---|---|---|---|---|---|---|
| | | Naive | Try 1x | Try $n$x | Naive | Try 1x | Try $n$x |
| gpt-4o | 12.5% | 327x | 0.87x | 0.46x | 5172x | 0.90x | 0.53x |
| o4-mini | 29.1% | 186x | 1.02x | 1.06x | 1265x | 1.06x | 1.22x |
| o3 | 35.2% | 161x | 1.08x | 1.28x | 480x | 1.13x | 1.47x |
| gpt-5 | 39.0% | 90x | 1.12x | 1.39x | 474x | 1.18x | 1.63x |

When incorporating time to review and redo work, the payoff from using a model shrinks. We do not include consideration of the time taken to review a human professional deliverable, although this would commonly occur for tasks in GDPval (either self-review of the professional's own work or review by a supervisor of a team member's work). We also do not include the possibility that the human deliverable is also undesirable. One further limitation of this analysis is that it does not capture the cost of catastrophic mistakes, which can be disproportionately expensive in some domains.

### A.2.2 WIN RATES BY SECTOR

We provide a sector-level breakdown of win rates in fig. 10. Results differ across industries. Some sectors have low win rates for all models, while in others (e.g., Government, Retail Trade, and Wholesale Trade), the strongest models approach parity on GDPval tasks.

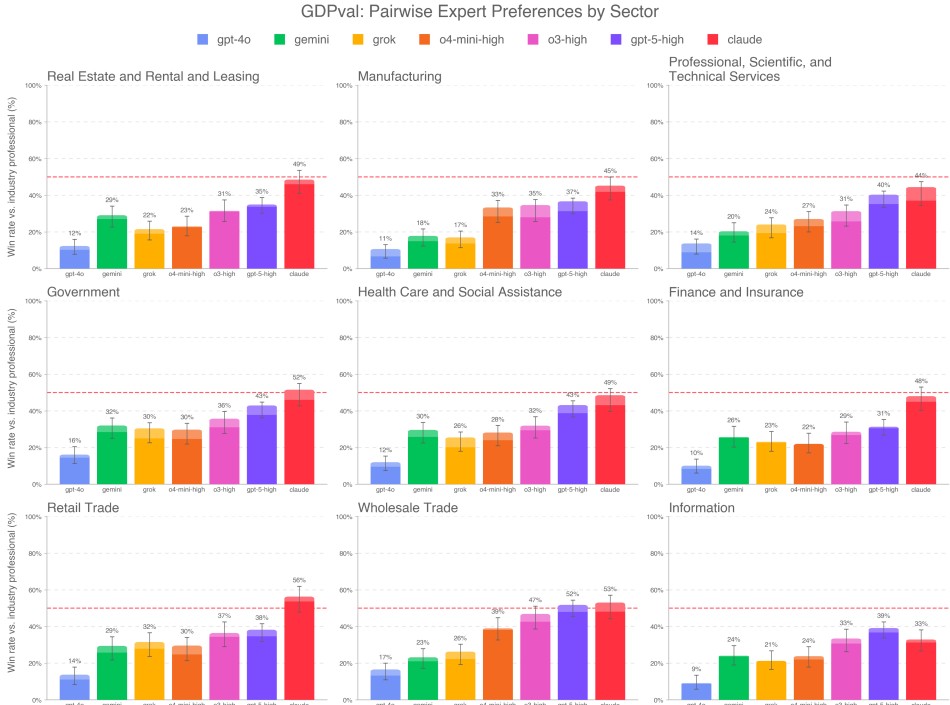

Figure 10: Win rate by sector

### A.2.3 WIN RATES BY OCCUPATION

We provide a detailed breakdown of win rates by occupation in fig. 11. Results vary: some occupations show consistently low win rates across all models, while others display near parity among multiple models.

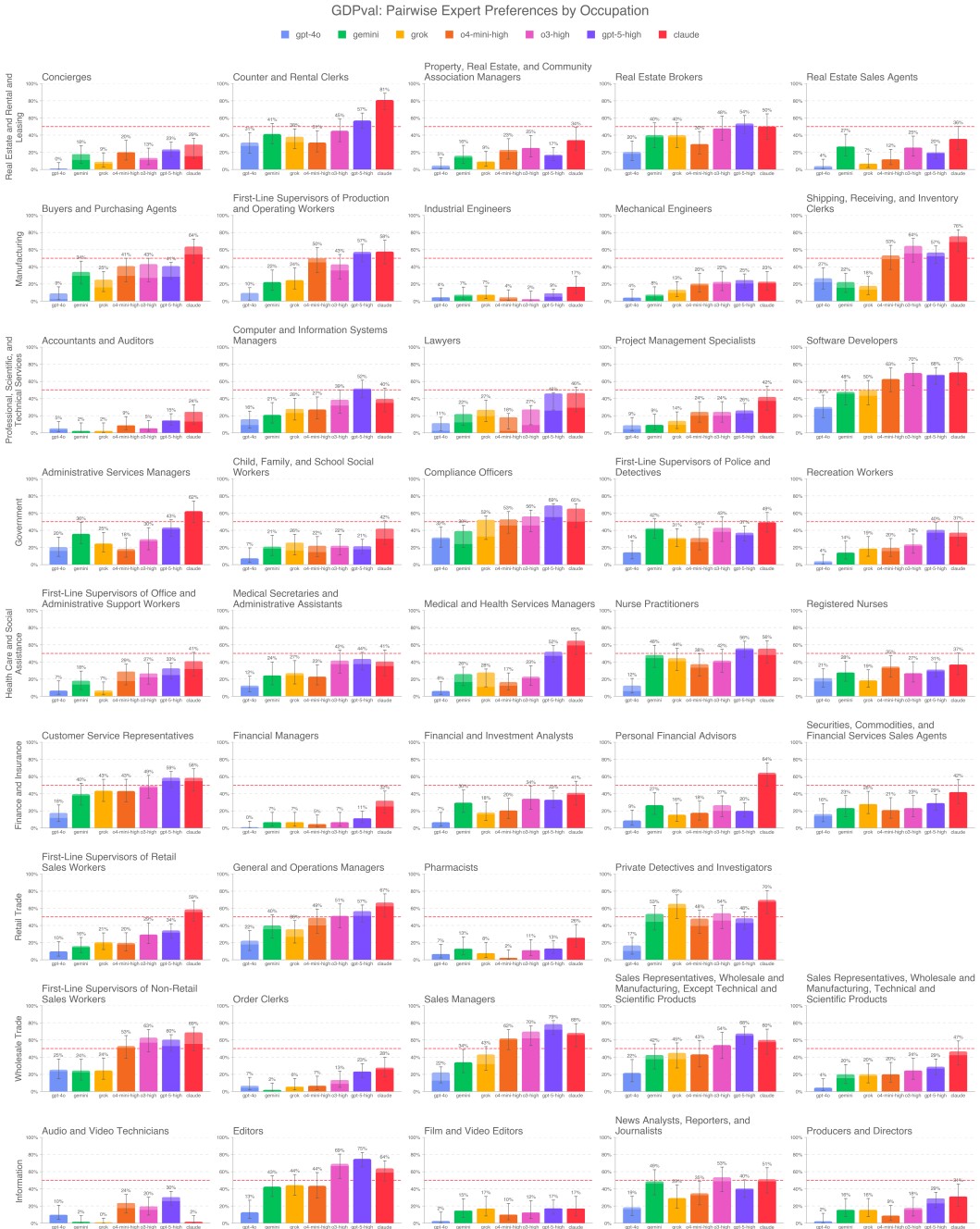

Figure 11: Win rate by occupation

### A.2.4 WIN RATES BY DELIVERABLE

We report win rates by deliverable type in fig. 12. Performance varies across formats, with Claude achieving the best results for all deliverables except pure text. GPT-5 high leads for pure text outputs, though overall win rates remain low.

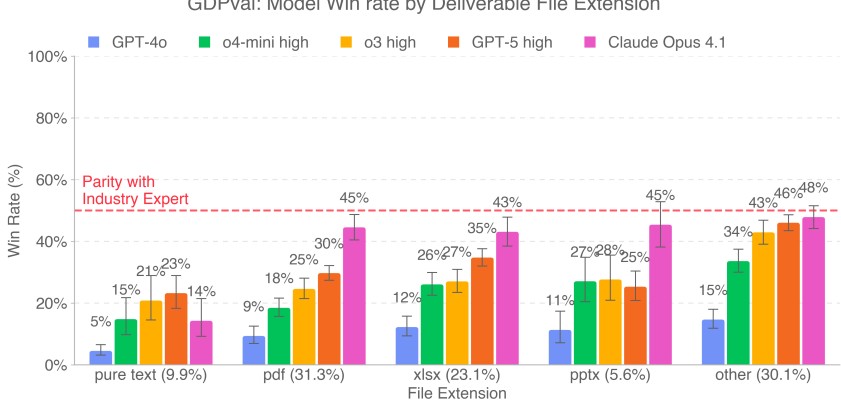

Figure 12: Win rate by deliverable file type

### A.2.5 WIN RATES BY TIME TO COMPLETE

We report win rates by task duration in fig. 13. Win rates are highest for shorter tasks (0–2 hours) and decline steadily as completion time increases. This indicates that models perform best on faster, less time-intensive tasks.

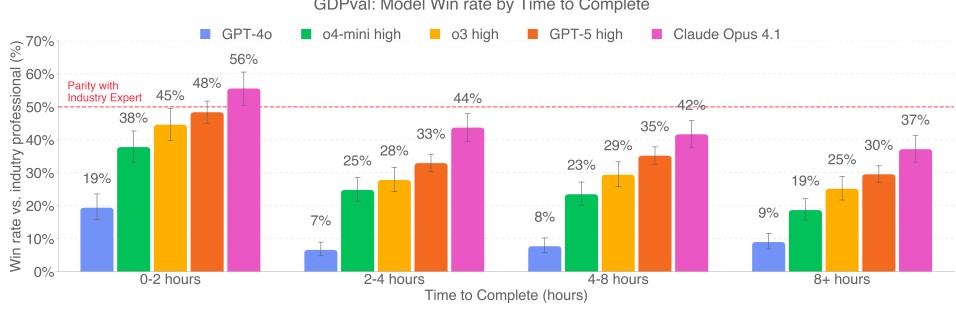

Figure 13: Win rate by time to complete task

### A.2.6 ADDITIONAL DETAIL ON MODEL FAILURES ANALYSIS

To generate this table, we analyzed the written justifications provided by expert graders when selecting a preferred completion in the head-to-head comparison. Through manual review, we observed that these explanations largely fell into three high-level categories: instruction following, factual accuracy, and multi-modal formatting. We then used a model to systematically cluster all grader justifications into these categories and further refine them into more specific failure subtypes (e.g., missing deliverables, wrong format, broken formatting). The resulting distribution provides a structured breakdown of where models most commonly fail, highlighting that while factual errors are relatively rare across all models, a substantial share of performance differences arise from instruction adherence and formatting quality, particularly in multi-modal outputs.

|  | Claude Opus 4.1 | Gemini 2.5 Pro | GPT-5-high | Grok 4 |
|---|---|---|---|---|
| **Factual errors** | | | | |
| Produces factual errors | 2.68% | 3.43% | 2.22% | 3.18% |
| **Instruction following errors** | | | | |
| Omits requested deliverables | 1.77% | 7.78% | 1.36% | 6.01% |
| Omits required content | 3.79% | 8.23% | 5.40% | 8.23% |
| Omits required visuals | 2.42% | 3.23% | 2.47% | 3.43% |
| Delivers wrong format | 0.56% | 4.19% | 0.81% | 4.90% |
| **Multi-modal formatting errors** | | | | |
| Provides unusable files | 0.96% | 1.36% | 0.81% | 0.86% |
| Broken formatting | 2.58% | 4.65% | 5.05% | 8.18% |

Table 2: Detailed model failure mode labels from our clustering of human expert preferences as a percent of all samples

We took the subset of GPT-5 model failures (tasks where the GPT-5 deliverable lost to the human expert), and then we asked other expert occupational graders to rate these subset samples as:

1. **Catastrophic:** The model completion would be catastrophic if used in real life because it is harmful or dangerously wrong (e.g., insulting a customer, giving the wrong diagnosis, recommending fraud, or suggesting actions that will cause physical harm).

2. **Bad:** The completion was bad and not fit for use, but not offensive or dangerous (e.g., rambling nonsense, completely irrelevant, or incoherent answers).

3. **Acceptable but subpar:** The completion was acceptable (and could be used) but the human produced a stronger response (e.g., model response lacked helpful detail compared to the human).

4. **N/A:** Disagree with original expert grader; the model completion was better than the human completion.

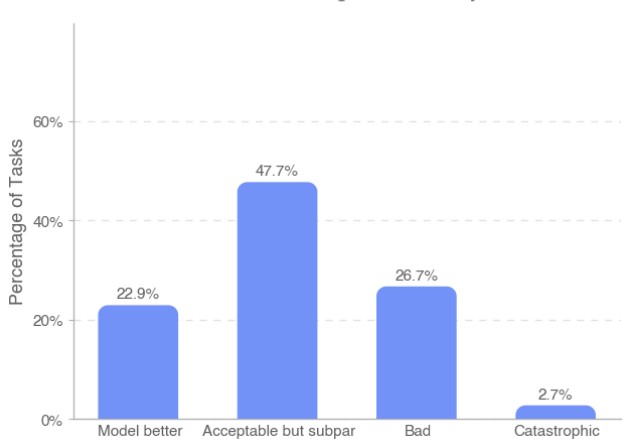

Figure 14: Experts rated GPT-5 model failures by categorized by severity of failure.

The most common categorization of a GPT-5 model failure was "acceptable but subpar." Another roughly 29% of ratings were for bad or catastrophic (with roughly 3% of failures marked as catastrophic). The 23% of ratings for "model better" roughly corresponds to the level of inter-rater agreement we observed in fig. 4b.

### A.2.7 UNDER-CONTEXTUALIZED GDPVAL

To assess how models handle task ambiguity, we created a modified version of GDPval with deliberately lower-context prompts. These shorter prompts omitted additional context such as where to locate specific data within reference files, how to approach the problem, or detailed formatting expectations for the final deliverable; the models had to "figure it out." On average, these revised prompts were 42% the length (by token count) of the original prompts.

This setting helped measure an aspect of professional knowledge work previously unaddressed in our evaluation: navigating ambiguity by figuring out what to work on and where to get the necessary inputs. We collected and graded GPT-5 completions with expert human graders and found the model's performance was worse on under-specified prompts. In particular, the models struggled to figure out context.

As a note: this experiment was run on an earlier version of the GDPval gold subset, and therefore the observed win rates do not match those in the main text of the paper.

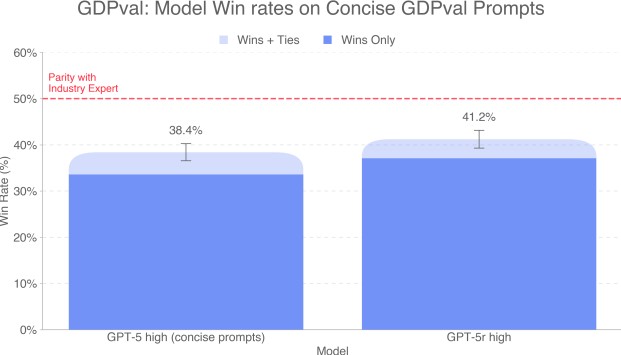

Figure 15: On the underspecified version of GDPval, GPT-5 performed worse as it struggled to figure out requisite context.

## A.3  ADDITIONAL DETAIL ON PROMPT-TUNING

Here is the prompt we give the agent to elicit capabilities (lightly edited to remove some specific details of our scaffolding setup).

---

**Prompt**

Special characters - Never use the character - (U+2011), since it will render poorly on some people's computers. Instead, always use - (U+002D) instead. - Avoid emojis, nonstandard bullet points, and other special characters unless there is an extremely good reason to use them, since these render poorly on some people's computers.

Graphics embedded within PDFs/slides - Make sure that any diagrams or plots are large enough to be legible (though not so large that they are ugly or cut off). In most cases they should be at least half the page width. - Plots and charts to visualize data are good. Simple graphics (like a flowchart with arrows) are good. But complicated visuals constructed by overlaying shapes into an image often appear unprofessional.

PDFs - Always use LibreOffice to create the PDF (it must be LibreOffice! If LibreOffice is not installed, you can install it yourself). Other libraries sometimes show weird artifacts on some computers.

Fonts - Always use fonts which are available across all platforms. We recommend Noto Sans / Noto Serif unless there is an extremely good reason to use something else. If you must use another font, embed the font in the pptx/word/etc doc.

Deliverable text - Do not link to submitted files in the deliverable text (links are not supported on the interface where these will be viewed). - Ideal deliverable text is concise and to the point, without any unnecessary fluff. 4 sentences max. - Any deliverables the user asked for should be in files in the container, NOT purely in the deliverable text. - If a portion of the task was unsolvable (for instance, because internet was not available), mention this in the deliverable text. - Your submission should be complete and self-contained. Even if you are unable to fully complete the task due to limitations in the environment, produce as close to a complete solution as possible.

Verbosity Always be clear and comprehensive, but avoid extra verbosity when possible.

Filetypes If the prompt does not request a specific filetype, use "standard" filetypes like PDF, PPTX, DOCX, XLSX, MP4, ZIP, etc.

Video files (mp4, mov) Extract a string of images from the video files and check the images to see whether the visual elements are corrupted.

Mandatory formatting checks Before you submit your deliverable, you MUST perform the following mandatory formatting checks. Take your time, do these thoroughly, they are extremely important!

STEP 1: Convert all visual deliverables to PNGs using LibreOffice. This includes pptx, docx, pdf, xlsx, etc. Convert it so that each page or slide is a separate PNG. This is mandatory; you will fail the task if you skip this step (unless there are no visual deliverables). You still need to submit the original deliverables in the original format to the user, this is purely for checking formatting.

STEP 2: Display the PNGs. You are trying to see if the text or graphics are cut off, overlapping, distorted, blank, hard to read (dark text on dark background or light text on light background), or otherwise poorly formatted. Look at each image thoroughly, zoom in if you need to see more closely. Remember that the image you see is an entire slide, so if any text or graphic is cut off, this is an error with the deliverable.

STEP 3: Programmatic formatting checks. For highly visual submissions (e.g. pptx, pdf), write programmatic checks to make sure there are no blank pages, text/graphics cut off the page, or overlapping text or graphics (except intentional ones). Also check that if there is a page or slide limit, it is respected.

STEP 4: Summarize the prompt's deliverable instructions, and match that to the portion of the deliverable that addresses it.

STEP 5: Right before submitting, check that the deliverables you have produced are exactly what you want to submit: deliverables should contain exactly the files you want to submit, with no extra files. Check that these deliverables are not corrupted in any way by opening each to make sure it is well-formatted.

If any of these checks reveal a formatting issue, fix them and go through steps 1-5 again. Take your time, be thorough, remember you can zoom in on details.

This is IMPORTANT and MANDATORY, go through each step one-by-one meticulously! Every formatting error is a MAJOR ISSUE THAT YOU NEED TO FIX! There is no time limit, be thorough, go slide by slide or page by page.

Finally – on the last line of your output text, add CONFIDENCE[XX], where XX is an integer between 0 and 100, inclusive, indicating your confidence that the submission is correct, follows instructions, and is well-formatted.

---

We performed best-of-N sampling by prompting a GPT-5 grader with the prompt, reference files, and deliverable files for four different submissions, then asking it to pick the best.

Table 3: Capabilities, file handling, and UI vs. API differences across models

| Capability | Claude Opus 4.1 | Gemini 2.5 Pro | Grok-4 | o4-mini | o3 | GPT-5 |
|---|---|---|---|---|---|---|
| **Input modalities** | Text, image | Text, image, audio, video | Text, image | Text, image | Text, image | Text, image |
| **Output modalities** | Text (audio via TTS/Live) | Text | Text | Text | Text | Text |
| **Context window** | 200k | 1M | 256k | 200k | 200k | 400k |
| **Max output tokens** | 32k | 64k | Not published | 100k | 100k | 128k |
| **Supported inputs** | .pdf, .txt, .json; images: .png, .jpg, .gif, .webp | .pdf, .docx, .pptx, .xlsx, .csv, .txt; images: .png, .jpg; audio: .mp3, .wav; video: .mp4 | *(obs.)* .pdf, .docx; images: .png, .jpg | .pdf, .docx, .pptx, .xlsx, .csv, .txt; images: .png, .jpg; .zip | Same as o4-mini | .pdf, .docx, .pptx, .xlsx, .csv, .txt; images: .png, .jpg |
| **Unsupported inputs** | .mp3, .wav, .mp4 (no direct audio/video) | (route-dependent) | *(obs.)* .mp3 rejected; .zip unreliable (docs: no official list) | .gdoc (export required) | .gdoc (export required) | .mp3, .wav (no direct audio files) |
| **Supported outputs** | .txt; via tools: .csv, .png, .pdf | .txt; via TTS/Live: .wav | .txt | .txt; via tools: .csv, .png, .pdf | .txt; via tools: .csv, .png, .pdf | .txt |
| **Max file size (typical)** | N/P; *(obs.)* ≤30MB/file; ≤20 files/chat | Apps: 100MB (non-video), 2GB (video); Files API: 2GB/file | Not published | 512MB/file (web/tools) | 512MB/file (web/tools) | 512MB/file (web/tools) |

*Notes.* "(obs.)" = behavior observed in GDPval tests. For Grok-4, vendor docs confirm *modalities=text+image* and *context=256k*. Claude processes PDFs >100 pages as text-only.

## A.4 ADDITIONAL TASK CHARACTERISTICS

Table 4: Summary statistics for GDPval gold subset tasks

| | **Mean** | **Std** | **Min** | **25%** | **50%** | **75%** | **Max** |
|---|---|---|---|---|---|---|---|
| Overall quality (1–5) | 4.47 | 0.32 | 3.18 | 4.30 | 4.50 | 4.70 | 5.00 |
| Difficulty (1–5) | 3.32 | 0.95 | 1.00 | 3.00 | 3.00 | 4.00 | 5.00 |
| Representativeness (1–5) | 4.50 | 0.74 | 2.00 | 4.00 | 5.00 | 5.00 | 5.00 |
| Avg time to complete (hrs) | 9.49 | 13.75 | 0.50 | 2.38 | 5.00 | 10.00 | 100.00 |
| Dollar value of task | $398.46 | $599.45 | $12.59 | $93.72 | $174.81 | $386.03 | $4,114.20 |

Table 5: Summary statistics for GDPval full set tasks

| | **Mean** | **Std** | **Min** | **25%** | **50%** | **75%** | **Max** |
|---|---|---|---|---|---|---|---|
| Overall quality (1–5) | 4.55 | 0.43 | 2.00 | 4.33 | 4.56 | 5.00 | 5.00 |
| Difficulty (1–5) | 3.20 | 0.92 | 1.00 | 3.00 | 3.00 | 4.00 | 5.00 |
| Representativeness (1–5) | 4.43 | 0.76 | 1.00 | 4.00 | 5.00 | 5.00 | 5.00 |
| Avg time to complete (hrs) | 8.63 | 24.70 | 0.25 | 2.00 | 4.00 | 8.00 | 605.00 |
| Dollar value of task | $391.44 | $1,296.67 | $8.53 | $70.70 | $147.31 | $354.12 | $32,028.70 |

### A.4.1 FILES AND ATTACHMENTS

Many traditional evaluations rely on text-in/text-out task formats. GDPval tasks incorporate a broad range of real-world file types (such as spreadsheets, documents, presentations, images, audio, video, and specialized formats like CAD). 67.7% of tasks required interaction with at least one reference file.

Table 6: File counts for GDPval gold set tasks

| | Mean | Std | Min | 25% | 50% | 75% | Max |
|---|---|---|---|---|---|---|---|
| Reference files | 1.92 | 3.47 | 0.00 | 0.00 | 1.00 | 2.00 | 38.00 |
| Deliverable files | 1.54 | 2.64 | 0.00 | 1.00 | 1.00 | 1.00 | 36.00 |

### A.4.2 O*NET TASKS, SKILLS, AND WORK ACTIVITIES

To ensure broad occupational representativeness, we analyzed the O*NET tasks, skills, and general work activities represented by GDPval tasks. The dataset covered 208 unique O*NET tasks, 25 occupational skills, and 26 work activities.

Most GDPval tasks involve multiple O*NET tasks, skills, and work activities.

Table 7: O*NET Tasks, Skills, and Work Activities coverage in gold set

| | Total unique in O*NET | Total in gold subset | Coverage (%) |
|---|---|---|---|
| O*NET Skills | 35 | 25 | 71.4% |
| O*NET Work Activities | 41 | 26 | 63.4% |
| O*NET Tasks | 1,470 | 208 | 14.15% |

### A.4.3 TASK SPECIFICATION

Occupational experts conducting human grading rated the specificity of instructions provided in each prompt. **89.07%** of tasks were rated as well-specified, indicating the instructions closely matched real-world expectations of clarity and detail.

Table 8: Task specification scores

| Label | %, gold set | %, full set |
|---|---|---|
| Underspecified | 8.28% | 8.41% |
| Well-specified | 89.07% | 89.34% |
| Overspecified | 2.66% | 2.26% |

### A.4.4 TASK REPRESENTATIVENESS

**Professional Services** *Qualification:* Technology and intellectual property attorney with partner roles at multiple AmLaw 100 firms in New York and California, and 15+ years of experience advising clients on emerging technologies, advertising, antitrust, and cross-border disputes and transactions.
*Quote:* Legal tasks included details that felt true to practice, like ambiguous fact patterns, disclosure of relevant legal considerations along with non-legal business goals, and realistic reference documents.

**Healthcare** *Qualification:* Nursing professional with 18+ years of expertise in emergency medicine, renal management, care coordination, and healthcare operations. Skilled in quality assurance, case management, and professional education.
*Quote:* These tasks captured the complexity of the role, requiring not only a keen ear for the physician's words, but also careful attention to clinical accuracy and professional formatting.

**Retail Trade** *Qualification:* Strategic retail executive with 15 years of experience growing prestige and niche beauty brands through national account leadership, $1B+ P&L ownership, and data-driven omnichannel strategies.
*Quote:* These tasks mirrored the work I performed regularly, including developing revenue forecasts, conducting competitive analysis, building executive-level presentations, and driving strategic initiatives for key retail partners within a global organization.

**Finance** *Qualification:* Fintech and Wall Street leader with 20+ years of experience in wealth management, asset management, and capital markets across global institutions and startups.

*Quote:* They reflected real-world scenarios that were nuanced and individualized, situations that only someone with years of experience in the field would fully comprehend. The language and details used in the tasks were directly drawn from actual industry practice, making them authentic and grounded in real-world application.

**Wholesale Trade** *Qualification:* National Accounts Sales Manager for US, China, and Sweden based brands/factories with over 25 years of experience selling to US based retailers.
*Quote:* All the tasks were in fact based upon real world tasks with back-up reference files and real-world data.

**Manufacturing** *Qualification:* Lead Industrial Engineer with 5+ years of experience managing large-scale projects and leading teams of 10+ engineers in industrial operations.
*Quote:* The redesign tasks stood out as especially true to real-world practice because they included specific design components and blocks, along with detailed drawings that incorporated precise measurements. They emphasized practical considerations such as visibility and optimizing walking distances to improve overall productivity, exactly the kind of detail-oriented focus that reflects actual engineering and operational priorities.

**Government** *Qualification:* Executive leader with 15+ years working at strategic and operational levels in government and non-profit sectors in housing, human service and labor market programs.
*Quote:* Many of the tasks demand the integration of multiple sources of information, nuanced decision-making, and tailored the work to varied audiences we serve in the workplace.

**Real Estate and Leasing** *Qualification:* Seasoned commercial real estate broker with 10 years of experience in investment sales, leasing, and managing real estate offices and agents.
*Quote:* The tasks capture the dynamics and expertise unique to specific sectors and settings.

**Information** *Qualification:* An experienced senior journalist and content leader with over 20 years in top-tier media, global corporations, and high-growth startups.
*Quote:* Most importantly, the tasks are anchored in real-world challenges and workplace goals. They push past obstacles, achieve workplace goals, and deliver real-world solutions and products.

**Additional Detail about Expert Qualifications** Less than 10% of applicants were selected to contribute tasks to our full set. The industry experts also brought occupational diversity, representing different company sizes, locations, and sub-specialties. Each occupation had a minimum of 5 qualified professionals.

Experts for each occupation were required to have previous experience in that specific occupation and sector based on the O*NET occupation definitions (of Labor Statistics, 2025).

A.5 FURTHER DETAIL ON TASK QUALITY CONTROL

A.5.1 MODEL-IN-THE-LOOP TASK REVIEW

We used OpenAI models to automatically screen each task submission across a variety of criteria and flag possible errors or omissions including: ensuring the task is relevant to the selected O*NET occupation, verifying the request involved tasks performed primarily on a computer, flagging if the task complexity was too simple (e.g., if the task seemed like 5 minutes of work instead of a longer-term piece of work), and indicating if there were no deliverable and reference files attached.

Because models can make mistakes, experts were instructed to take model feedback as a suggestion rather than a direction. Experts retained final responsibility for task accuracy and completeness; the model did not autonomously alter tasks.

A.5.2 HUMAN EXPERT REVIEWERS

Human reviewers conducted multiple rounds of review on each task. Reviewers were primarily sourced from the original expert pool based on demonstrated excellence in task creation. Initially, our researchers manually reviewed all tasks to identify experts who produced consistently high-quality tasks; these individuals were trained and promoted to reviewers. The most skilled reviewers were further trained to become lead reviewers, responsible for identifying, mentoring, and promoting additional qualified reviewers from within the expert pool. Throughout the review process, the

research team regularly performed quality-control checks on tasks signed off by reviewers, ensuring ongoing alignment and quality standards.

### A.5.3 ITERATIVE REVIEW PROCESS

The iterative review process included at least the following 3 stages:

1. **Generalist initial review**: A generalist reviewer confirmed the task adhered to project requirements.

2. **Occupation-specific expert review**: An occupation-specific reviewer assessed the representativeness of the task for the occupation, and confirmed that the task was possible for another member of the occupation to complete with the provided context.

3. **Final iterative reviewer feedback loop**: A third expert reviewer provided iterative feedback and worked with experts until the task met our rigorous quality standards.

Prior to the main review pipeline, all task writers completed a "sandbox" task. After an initial onboarding survey confirming that they understood campaign requirements, writers created a first task in the "Sandbox" and received iterative feedback from an experienced reviewer. A writer could begin contributing to the live campaign only once their sandbox task was signed off as meeting our quality and representativeness standards.

### A.6 HUMAN BASELINER RECRUITMENT, TRAINING, AND QUALITY CONTROL

This section provides additional detail on the recruitment, compensation, training, and evaluation procedures for human baseliners and occupational experts, following recommended reporting practices (Handa et al., 2025).

We recruited expert industry professionals to create realistic tasks based on their professional work experience. Experts were required to have a minimum of 4 years of professional experience in their occupation and a strong resume with a demonstrated history of professional recognition, promotion, and management responsibilities. The average expert had 14 years of experience. We further required experts to pass a video interview, a background check, a training and a quiz to participate in the project. Experts were well compensated for their time and experience.

### A.6.1 RECRUITMENT AND POPULATION CRITERIA

Human baseliners and occupational experts were recruited for each occupation through postings on platforms like LinkedIn and targeted outreach. Applicants were screened using their resumes and LinkedIn, an AI-assisted interview, and, for graders, an occupation-specific questionnaire. Eligibility required at least 4 years of full-time experience in the target occupation and fluency in English (all tasks were written and graded in English). Approximately 10% of applicants met these criteria and were selected; 90% were excluded based on insufficient experience. Experts were predominantly based in the United States, United Kingdom, and Canada.

Demographic information on baseliners was not collected beyond region and prior work experience. Therefore, full demographic reporting is not available.

No author served as a baseliner or grader.

### A.6.2 COMPENSATION

All human contributors were compensated at professionally competitive hourly rates for their occupation, experience level, and geographic region. Rates varied by occupation and resume strength, but all contributors were paid well above U.S. minimum professional contracting wages, with a minimum hourly rate of $40/hr. Task writers and graders were compensated at the same levels during training, and performance-based bonuses were occasionally offered to increase throughput subject to quality checks.

### A.6.3 TRAINING PROCEDURES

All experts who created tasks participated in live training calls covering grading standards, exemplar tasks, common errors, and expectations around deliverable quality. Contributors also had access to office hours and dedicated Slack channels for ongoing clarification and support throughout task writing and grading. Writers and graders were compensated for training time.

### A.6.4 EXECUTION AND QUALITY CONTROL

Experts determined the number of tasks they completed per session (instrument length was not fixed). Items were not randomized.

Quality control was performed throughout data collection. Our staff monitored Average Handling Time (AHT), and reviewers flagged low-effort or low-quality submissions. Tasks failing quality thresholds were excluded from the gold set. For more on the review pipeline, see appendix A.5.

Humans and models did not use equivalent UIs: humans worked through a graphical interface, whereas models interacted through APIs or first-party UIs depending on file-handling reliability.

### A.7 AUTOMATED GRADER DETAILS

### A.7.1 AUTOMATED GRADER CONSENSUS METRICS

To measure automated grader performance, we measured the agreement rate between scores given by the automated grader vs. human expert graders for the same sample. We also compared grading agreement between human experts who had graded the same sample.

**Human-automated grader Agreement.** For a given sample $s$, let the human score $H$ and automated grader score $A$ take values in $\{0, 0.5, 1\}$, where $1$ indicates preference for the model deliverable, $0$ indicates preference for the human deliverable, and $0.5$ indicates a tie. The agreement between human and automated grader is defined as

$$A_s^{\mathrm{HA}} = \mathbb{E}\big[1 - |H - A|\big].$$

The model-level human–automated grader agreement is the mean of $A_s^{\mathrm{HA}}$ over all samples for that model.

**Human Inter-Rater Agreement.** For a given sample $s$, let the human scores $H_1$ and $H_2$ take values in $p \in \{0, 0.5, 1\}$. We measure human inter-rater agreement as the following expectation over two randomly sampled human ratings

$$A_s^{\mathrm{HH}} = \mathbb{E}\big[1 - |H_1 - H_2|\big].$$

For a given sample, we estimate this quantity by the empirical mean over all pairs of ratings for that sample. The final human inter-rater agreement for a model is the mean of these sample-level scores over all samples with at least two human graders. Existing grader inter-reliability statistics such as Cohen's kappa, Fleiss' kappa, and Krippendorff's alpha are less directly applicable here, since our graders output ordinal scores in $\{0, 0.5, 1\}$.

### A.7.2 AUTOMATED GRADER CORRELATION RESULTS

Over three automated grader sweeps on our dataset[9], average human-automated grader agreement was 65.7% and human inter-rater agreement was 70.8%. Plots below show 95% confidence intervals obtained by bootstrapping (resampling with replacement the available automated grader scores or human grades for each sample, computing the mean per sample, and averaging across all samples or for the specified model).

---

[9]Metrics were calculated over all samples where the automated grader did not encounter systems errors and returned a valid score. We also excluded 12 tasks (out of the 220 in our open-sourced eval set) that the automated grader frequently could not grade or was less likely to grade reliably due to its limitations, described later.

Our automated grader, based on GPT-5-high, shows lower correlation with human expert graders when assessing outputs from capable OpenAI models. This aligns with empirical evidence that models often favor their own responses Panickssery et al. (2024). Both agreement metrics are highest for less capable models, since their outputs are easier to distinguish from human deliverables and are less likely to be preferred.

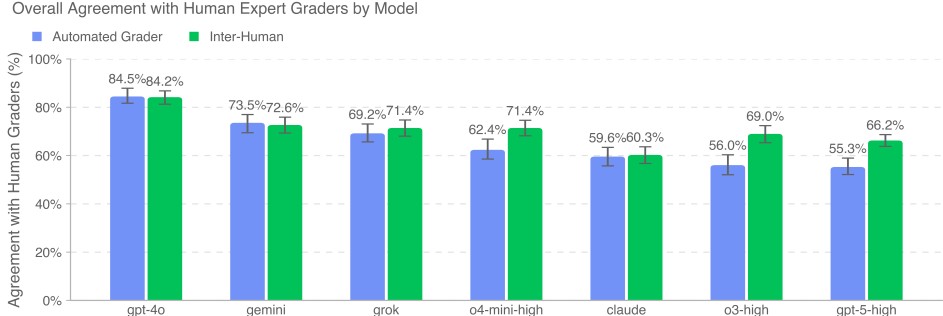

Figure 16: Average human-automated grader agreement is most closely aligned with human inter-rater agreement for non-OpenAI models. Both agreement metrics are highest for less capable models, as they can be more frequently distinguished from human deliverables and are less likely to be chosen.

### A.7.3 AUTOMATED GRADER RUBRICS

Rubrics used by the autograder are first written by experts trained in rubric writing, then reviewed by another human expert from the same occupation as the original task writer to improve quality. We iteratively improved these rubrics using a model prompted for rubric refinement, and every revision was checked again by an expert rubric writer. This process helped ensure that the rubrics were clear, consistent, and aligned with domain expectations.

### A.7.4 AUTOMATED GRADER LIMITATIONS

In the open-source set we mark 12 out of 220 tasks as ungradable due to limitations of the automated grader.

1. **Internet Access:** Tasks which strictly require internet (e.g., tasks that ask agents to find music online and download it) are not possible to grade because the grader does not have internet access.

2. **Python**: The automated grader operates in a container that only allows for running Python. Because of this, we excluded 3 Software Developers tasks that require running other languages and downloading external dependencies to properly test.

3. **Font Packages** Although the automated grader has most metrically-identical fonts (e.g., Liberation Sans instead of Arial), some font packages used in human deliverables still causes certain deliverables to be rendered differently than they would appear on a computer that has these fonts installed.

4. **Speech-to-text transcription:** The automated grader has limited speech to text functionality inside the container, and struggles with non-voice sounds.

### A.7.5 AUTOMATED GRADER PACKAGES

To ensure the model can process a wide variety of file types in `GDPval`, the following packages are pre-installed in the base production Docker image. These were also made available to the agent during sampling of OpenAI models.

```
jupyter-client==8.6.1          jupyterlab==4.1.8
jupyter-core==5.5.1            jupyterlab-pygments==0.3.0
jupyter-server==2.14.0         jupyterlab-server==2.27.1
```

```
aiohttp==3.9.5                          torchvision==0.20.1
hypercorn==0.14.3                       PyMuPDF==1.21.1
notebook==6.5.1                         pdf2image==1.16.3
nbclassic==0.4.5                        pyth3==0.7
pydantic==1.10.2                        h5py==3.8.0
fastapi[all]==0.95.2                    tables==3.8.0
websockets==10.3                        rarfile==4.0
tqdm==4.64.0                            odfpy==1.4.1
matplotlib==3.6.3                       pymc==4.0.1
matplotlib-venn==0.11.6                 jax==0.2.28
numpy==1.24.0                           pyxlsb==1.0.8
numpy-financial==1.0.0                  keras==2.6.0
scipy==1.14.1                           xgboost==1.4.2
pandas==1.5.3                           loguru==0.5.3
statsmodels==0.13.5                     plotly==5.3.0
sympy==1.13.1                           graphviz==0.17
seaborn==0.11.2                         fuzzywuzzy==0.18.0
scikit-learn==1.1.3                     pydot==1.4.2
nltk==3.9.1                             gensim==4.3.1
plotnine==0.10.1                        pypandoc==1.6.3
shapely==1.7.1                          einops==0.3.2
fiona==1.9.2                            reportlab==3.6.12
geopandas==0.10.2                       gradio==2.2.15
ffmpeg-python==0.2.0                    mutagen==1.45.1
pydub==0.25.1                           librosa==0.8.1
moviepy==1.0.3                          svglib==1.1.0
opencv-python==4.5.5.62                 gtts==2.2.3
Pillow==9.1.0                           textblob==0.15.3
python-docx==0.8.11                     rasterio==1.3.3
python-pptx==0.6.21                     rdflib==6.0.0
openpyxl==3.0.10                        rdkit==2024.9.6
xml-python==0.4.3                       biopython==1.84
geopy==2.2.0                            cairosvg==2.5.2
scikit-image==0.20.0                    markdownify==0.9.3
folium==0.12.1                          anytree==2.8.0
wordcloud==1.9.2                        pdfplumber==0.6.2
faker==8.13.2                           trimesh==3.9.29
fpdf2==2.8.3                            svgwrite==1.4.1
soundfile==0.10.2                       pdfrw==0.4
kerykeion==2.1.16                       pyzbar==0.1.8
pdfkit==0.6.1                           dlib==19.24.2
pycountry==20.7.3                       mtcnn==0.1.1
countryinfo==0.1.2                      imgkit==1.2.2
tabulate==0.9.0                         chardet==3.0.4
shap==0.39.0                            bokeh==2.4.0
pylog==1.1                              tabula==1.0.5
pyprover==0.5.6                         camelot-py==0.10.1
pytesseract==0.3.8                      exchange_calendars==3.4
qrcode==7.3                             weasyprint==53.3
basemap==1.3.9                          pronouncing==0.2.0
pygraphviz==1.7                         cryptography==3.4.8
networkx==2.8.8                         spacy==3.4.4
pyttsx3==2.90                           requests==2.31.0
nashpy==0.0.35                          mne==0.23.4
docx2txt==0.8                           pyopenssl==21.0.0
typing-extensions==4.10.0               snowflake-connector-python==2.7.12
torch==2.5.1                            databricks-sql-connector==0.9.1
torchaudio==2.5.1                       ddtrace~=2.8.1
torchtext==0.18.0                       datadog~=0.49.1
```

```
pytest~=8.2.0                 catboost~=1.2.7
pytest-cov~=5.0.0             lightgbm~=4.5.0
pytest-json-report~=1.5.0     imblearn~=0.0
coverage~=7.5.1               imbalanced-learn~=0.12.3
pytest-asyncio~=0.23.6        rapidfuzz~=3.10.1
```

We also installed the following additional packages, and we tell the model in the prompt it has access to these additional packages:

```
libreoffice
aspose-words==25.8.0
av==11.0.0
cadquery==2.4.0
cadquery-ocp==7.7.0
pedalboard==0.9.9
pyloudnorm==0.1.1
srt==3.5.3
xlrd==2.0.1
```

## A.8 FURTHER METHODOLOGICAL DETAILS ON SELECTING OCCUPATIONS

### A.8.1 ASSIGNING OCCUPATIONS TO SECTORS.

We assigned occupations to sectors by using the 2023 BLS National Employment Matrix from of Labor Statistics (2025) to identify the sector with the highest employment for each occupation. This involved filtering to "Line Item" occupations, taking the first two digits of NAICS codes, dropping "total employment" rows, summing 2023 employment, and assigning each occupation to the sector with the largest share of employment.

| Sector | % GDP | Top Occupations and Total Compensation (in Billions USD) |
|---|---|---|
| Real Estate and Rental and Leasing | 13.8% | **Property/RE/Community Association Managers — $24.54B**

Counter and Rental Clerks — $17.42B
Real Estate Sales Agents — $13.53B
Real Estate Brokers — $4.55B
Concierges — $1.80B |
| Manufacturing | 10.0% | First-Line Supervisors of Production and Operating Workers — $51.07B
Buyers and Purchasing Agents — $39.79B
Shipping, Receiving, and Inventory Clerks — $38.50B
Industrial Engineers — $37.79B
Mechanical Engineers — $31.57B |
| Professional, Scientific, and Technical Services | 8.1% | Software Developers — $239.18B

Lawyers — $136.66B
Accountants and Auditors — $135.44B
Computer and Information Systems Managers — $121.44B
Project Management Specialists — $108.77B |
| Government | 11.3% | Compliance Officers — $33.80B
Administrative Services Managers — $32.03B
Child, Family, and School Social Workers — $24.10B
First-Line Supervisors of Police and Detectives — $17.00B
Recreation Workers — $11.51B |
| Health Care and Social Assistance | 7.6% | Registered Nurses — $323.05B

First-Line Supervisors of Office/Admin Support — $107.02B
Medical & Health Services Managers — $77.93B
Nurse Practitioners — $40.58B
Medical Secretaries & Admin Assistants — $37.87B |
| Finance and Insurance | 7.4% | Financial Managers — $147.74B
Customer Service Representatives — $123.70B
Securities, Commodities, and Financial Services Sales Agents — $52.14B
Personal Financial Advisors — $43.33B
Financial and Investment Analysts — $39.67B |
| Retail Trade | 6.3% | General & Operations Managers — $477.16B
1st-Line Supervisors of Retail Sales Workers — $58.27B
Pharmacists — $45.12B
Private Detectives & Investigators — $2.39B |
| Wholesale Trade | 5.8% | Sales Reps, Wholesale & Mfg (Except Tech/Scientific) — $103.21B
Sales Managers — $97.16B
Sales Reps, Wholesale & Mfg (Tech/Scientific) — $33.66B
1st-Line Supervisors of Non-Retail Sales Workers — $21.43B
Order Clerks — $3.86B |
| Information | 5.4% | Producers & Directors — $16.60B
Editors — $8.18B
News Analysts, Reporters, and Journalists — $4.41B
Audio & Video Technicians — $4.30B
Film & Video Editors — $2.41B |

Table 9: Sectors, their value added as a percentage of U.S. GDP (Q2 2024), with representative top occupations and total compensation in billions (USD).

### A.8.2 DETAIL ABOUT O*NET DATA SOURCE

**Occupations in GDPval.** We arrived at 831 occupations by filtering to "Detailed" occupations from the May 2024 OEWS national employment and wage statistics U.S. Bureau of Labor Statistics (2025) to exclude any aggregate employment categories. We dropped "All Other" occupations,

which are catch-all categories within a broader group that bundle together occupations that don't fit into any of the detailed occupations in that group. Dropping "All Other" occupations left us with 761 occupations.

### A.8.3 Calculation of Total Wages Earned by Occupation.

Estimated total wages earned is calculated as total employment * mean annual salary for jobs with annual salaries, and total employment * hourly salary * typical work year of 2080 hours for jobs with only hourly salaries. The determination of which jobs had annual vs. hourly salaries was included in O*NET data. 2080 hours is cited as a "typical work year" by the Bureau of Labor Statistics (BLS), assuming someone works 40 hours per week. This is an imperfect estimate (eg., the BLS acknowledges actors "generally do not work 40 hours per week, year round") but is the most precise estimate provided by the BLS.

### A.8.4 Computing Digital Scores for Occupations.

To classify occupations as predominantly digital, we use a task-based approach. For many occupations, the O*NET database contains task statements and ratings that list all the tasks performed by a worker in an occupation.[10] The O*NET data is provided on the 6-digit SOC occupational code level (SOC-6). We map the O*NET SOC-6 occupations and the corresponding tasks to occupations in the OEWS dataset which reports wages at the 4-digit SOC level ("SOC-4"). For each SOC-4 occupation, we classify its tasks as either digital or non-digital using a prompted GPT-4o model that receives both the occupation and task and the prompt, "For each Task, Occupation, and Sector, classify the task as primarily manual/physical (Return 0) or primarily digital/knowledge work (Return 1). Reply with ONLY 0 or 1. 0 = manual; 1 = digital." We then calculate the weighted share of digital tasks for each occupation. Occupations are classified as digital if their digital share exceeds a threshold of 0.60.

To calculate the weights for our weighted task share, we use task ratings data from O*NET surveys, which includes the relevance, frequency, and importance of each task of the occupation.[11] We first calculate an Adjusted Task Score for each combination of 6-digit SOC occupation and task. This score is defined as the simple average of the three normalized task ratings: task frequency, task importance, and task relevance. Each rating is normalized relative to the maximum observed rating (e.g. the importance ratings are out of 5).[12] If one of these ratings is missing for a task, we impute the value with the mean of that rating across all tasks within the same occupation. For example, if a task lacks a frequency rating, we assign it the average normalized frequency rating of all tasks in the occupation.

We then aggregate these 6-digit Adjusted Task Scores into 4-digit Adjusted Task Scores (for each set of 4-digit SOC occupations and tasks). We do this by summing the SOC-6 Adjusted Task Scores of SOC-6 occupations within a SOC-4 occupation for each task.[13] For example, the SOC-4 occupation Computer Occupations, All Other combines two 6-digit SOC occupations (*Information Security Engineers* and *Penetration Testers*) which have one task in common: "Identify security system weaknesses, using penetration tests." This task has two SOC-6 Adjusted Task Scores which are added together to create the SOC-4 Adjusted Task Score.

Next, we calculate the Weighted Task Share for each combination of 4-digit SOC occupation and task. The Weighted Task Share is the Adjusted Task Score of the occupation-task pair divided by the sum of Adjusted Task Scores of that occupation. For each occupation, the sum of Weighted Task

---

[10]Note that while O*NET distinguishes between Core and Supplemental tasks in its task data, we treat these two task types equally in our calculation of task share.

[11]For the two occupations without O*NET 28.3 task ratings ("Facilities Managers" and "Medical Dosimetrists"), we used task ratings from O*NET 29.0.

[12]The maximum importance value is 5, and the maximum relevance value is 100. The maximum frequency value is 7, and the values correspond to categories (e.g. 1="yearly or less," 7="hourly or more"). We treat these frequency ratings as a proxy for relative task importance. While the scale is categorical rather than interval, a rating of 7 unambiguously indicates that a task is performed more often and is typically more central to day-to-day work than a task rated 1.

[13]If a SOC-4 occupation is mapped to one SOC-6 occupation, the SOC-6 and SOC-4 Adjusted Task Scores are the same.

Share across all its tasks is equal to one. The Weighted Task Share gives us a measure of the relative significance of each task for a given occupation. These Weighted Task Shares are the weights used to calculate the weighted share of digital tasks for each occupation.

Table 10: Tasks and Digital Score for Counter and Rental Clerks (SOC 41-2021)

| SOC Code | Occupation Title | Task | Weighted Task Share | Digital Classification | WTS × Digital | Occupation Digital Score |
|---|---|---|---|---|---|---|
| 41-2021 | Counter and Rental Clerks | Compute charges for merchandise or services and receive payments. | 6.97% | 1 | 6.97% | |
| 41-2021 | Counter and Rental Clerks | Receive orders for services, such as rentals, repairs, dry cleaning, and storage. | 6.67% | 1 | 6.67% | |
| 41-2021 | Counter and Rental Clerks | Explain rental fees, policies, and procedures. | 6.53% | 1 | 6.53% | |
| 41-2021 | Counter and Rental Clerks | Provide information about rental items, such as availability, operation, or description. | 6.56% | 1 | 6.56% | |
| 41-2021 | Counter and Rental Clerks | Advise customers on use and care of merchandise. | 6.52% | 1 | 6.52% | |
| 41-2021 | Counter and Rental Clerks | Greet customers and discuss the type, quality, and quantity of merchandise sought for rental. | 6.65% | 0 | 0.00% | |
| 41-2021 | Counter and Rental Clerks | Answer telephones to provide information and receive orders. | 6.62% | 1 | 6.62% | |
| 41-2021 | Counter and Rental Clerks | Inspect and adjust rental items to meet needs of customer. | 6.38% | 0 | 0.00% | |
| 41-2021 | Counter and Rental Clerks | Prepare rental forms, obtaining customer signature and other information, such as required licenses. | 6.23% | 1 | 6.23% | |
| 41-2021 | Counter and Rental Clerks | Rent items, arrange for provision of services to customers, and accept returns. | 6.39% | 0 | 0.00% | |
| 41-2021 | Counter and Rental Clerks | Keep records of transactions and of the number of customers entering an establishment. | 6.20% | 1 | 6.20% | |
| 41-2021 | Counter and Rental Clerks | Receive, examine, and tag articles to be altered, cleaned, stored, or repaired. | 6.27% | 0 | 0.00% | |
| 41-2021 | Counter and Rental Clerks | Reserve items for requested times and keep records of items rented. | 5.89% | 1 | 5.89% | |
| 41-2021 | Counter and Rental Clerks | Prepare merchandise for display or for purchase or rental. | 5.65% | 0 | 0.00% | |
| 41-2021 | Counter and Rental Clerks | Recommend and provide advice on a wide variety of products and services. | 5.67% | 1 | 5.67% | |
| 41-2021 | Counter and Rental Clerks | Allocate equipment to participants in sporting events or recreational activities. | 4.81% | 0 | 0.00% | |
| | | | 100.00% | | 63.86% | 64% |

Table 11: Occupation Digital Scores

| Sector | Occupation | Digital Score |
|---|---|---|
| Finance and Insurance | Financial Managers | 0.984 |
| Finance and Insurance | Customer Service Representatives | 1.000 |
| Finance and Insurance | Securities, Commodities, and Financial Services Sales Agents | 1.000 |
| Finance and Insurance | Personal Financial Advisors | 1.000 |
| Finance and Insurance | Financial and Investment Analysts | 0.962 |
| Government | Compliance Officers | 0.849 |
| Government | Administrative Services Managers | 0.93 |
| Government | Child, Family, and School Social Workers | 0.92 |
| Government | First-Line Supervisors of Police and Detectives | 0.76 |
| Government | Recreation Workers | 0.625 |
| Health Care and Social Assistance | Registered Nurses | 0.810 |
| Health Care and Social Assistance | First-Line Supervisors of Office and Administrative Support Workers | 0.966 |
| Health Care and Social Assistance | Medical and Health Services Managers | .95 |
| Health Care and Social Assistance | Nurse Practitioners | 0.926 |
| Health Care and Social Assistance | Medical Secretaries and Administrative Assistants | 0.93 |
| Information | Producers and Directors | 0.912 |
| Information | Editors | 1.000 |
| Information | News Analysts, Reporters, and Journalists | 0.967 |
| Information | Audio and Video Technicians | 0.793 |
| Information | Film and Video Editors | 1.000 |
| Manufacturing | First-Line Supervisors of Production and Operating Workers | 0.79 |
| Manufacturing | Buyers and Purchasing Agents | 0.850 |
| Manufacturing | Shipping, Receiving, and Inventory Clerks | 0.73 |
| Manufacturing | Industrial Engineers | 0.93 |
| Manufacturing | Mechanical Engineers | 0.949 |
| Professional, Scientific, and Technical Services | Software Developers | 1.000 |
| Professional, Scientific, and Technical Services | Lawyers | 1.000 |
| Professional, Scientific, and Technical Services | Accountants and Auditors | 1.000 |
| Professional, Scientific, and Technical Services | Computer and Information Systems Managers | 0.95 |
| Professional, Scientific, and Technical Services | Project Management Specialists | 1.000 |
| Real Estate and Rental and Leasing | Property, Real Estate, and Community Association Managers | 0.852 |
| Real Estate and Rental and Leasing | Counter and Rental Clerks | 0.638 |
| Real Estate and Rental and Leasing | Real Estate Sales Agents | 0.879 |
| Real Estate and Rental and Leasing | Real Estate Brokers | 1.000 |
| Real Estate and Rental and Leasing | Concierges | 0.71 |
| Retail Trade | General and Operations Managers | 0.93 |
| Retail Trade | First-Line Supervisors of Retail Sales Workers | 0.75 |
| Retail Trade | Pharmacists | 0.905 |
| Retail Trade | Private Detectives and Investigators | 0.75 |
| Wholesale Trade | Sales Representatives, Wholesale and Manufacturing, Except Technical and Scientific Products | 0.80 |
| Wholesale Trade | Sales Managers | 0.95 |
| Wholesale Trade | Sales Representatives, Wholesale and Manufacturing, Technical and Scientific Products | 0.857 |
| Wholesale Trade | First-Line Supervisors of Non-Retail Sales Workers | 0.77 |
| Wholesale Trade | Order Clerks | 0.94 |

### A.8.5 HANDLING MISSING DATA.

1. **Missing Task Statements.** Some occupations in OEWS lacked associated task statements or ratings. Forty-seven of these were broad "All Other" categories without component tasks

[14]; twelve others were split into finer sub-occupations in O*NET 29.0 (as of August 2025). For the latter, we incorporated the full set of component tasks from their sub-occupations in O*NET 29.0. The exact reconciliation of how we mapped these 12 occupations is below. We distinguish between SOC-6 and SOC-4 because these levels meaningfully differ in granularity. SOC-6 (O*NET-SOC) occupations have task-level ratings, whereas many SOC-4 (BLS SOC) aggregates do not have their own task lists and must be constructed by combining constituent SOC-6 occupations. We used the Bureau of Labor Statistics-provided SOC-6 to SOC-4 crosswalk to perform all aggregations.

(a) **Tour and Travel Guides:** This SOC Code is broken out into two occupations: Tour Guides and Escorts and "Travel Guides". We added the tasks from both occupations.

(b) **Miscellaneous Construction and Related Workers:** This SOC Code is broken out into three occupations: "Segmental Pavers", "Weatherization Installers and Technicians", and "Construction and Related Workers, All Other". We added all of the tasks from "Segmental Pavers" and "Weatherization Installers." "Construction and Related Workers, All Other" is a general occupation category without component tasks.

(c) **Teaching Assistants:** This SOC Code is broken out into three occupations: Teaching Assistants, Preschool, Elementary, Middle, and Secondary School, Except Special Education, Teaching Assistants, Special Education, and Teaching Assistants, All Other. We added the tasks from Teaching Assistants, Preschool, Elementary, Middle, and Secondary School, Except Special Education, and Teaching Assistants, Special Education. Teaching Assistants, All Other is a general occupation category without component tasks.

(d) **Buyers and Purchasing Agents:** This SOC Code is broken out into three occupations: Buyers and Purchasing Agents, Farm Products, Wholesale and Retail Buyers, Except Farm Products, and Purchasing Agents, Except Wholesale, Retail, and Farm Products. We added the tasks from the three occupations.

(e) **Substance Abuse, Behavioral Disorder, and Mental Health Counselors:** This SOC Code is broken out into two occupations: Substance Abuse and Behavioral Disorder Counselors and Mental Health Counselors. We added the tasks from both occupations.

(f) **Clinical Laboratory Technologists and Technicians:** This SOC Code is broken out into six occupations: Medical and Clinical Laboratory Technologists, Cytogenetic Technologists, Cytotechnologists, Histotechnologists, Medical and Clinical Laboratory Technicians, and Histology Technicians. We added the tasks from all these occupations.

(g) **Special Education Teachers, Kindergarten and Elementary School:** This SOC Code is broken out into two occupations: Special Education Teachers, Kindergarten and Special Education Teachers, Elementary School. We added the tasks from both occupations.

---

[14]These occupations were: Entertainers and Performers, Sports and Related Workers, All Other; Postsecondary Teachers, All Other; Production Workers, All Other; Office and Administrative Support Workers, All Other; Teachers and Instructors, All Other; Surgeons, All Other; Information and Record Clerks, All Other; Community and Social Service Specialists, All Other; Educational Instruction and Library Workers, All Other; Sales and Related Workers, All Other; Education Administrators, All Other; Social Workers, All Other; Legal Support Workers, All Other; Food Preparation and Serving Related Workers, All Other; Personal Care and Service Workers, All Other; Food Processing Workers, All Other; Motor Vehicle Operators, All Other; Financial Clerks, All Other; Media and Communication Workers, All Other; Counselors, All Other; Social Sciences Teachers, Postsecondary, All Other; First-Line Supervisors of Protective Service Workers, All Other; Dentists, All Other Specialists; Material Moving Workers, All Other; Helpers, Construction Trades, All Other; Drafters, All Other; Media and Communication Equipment Workers, All Other; Metal Workers and Plastic Workers, All Other; Cooks, All Other; Designers, All Other; Life Scientists, All Other; Building Cleaning Workers, All Other; Precision Instrument and Equipment Repairers, All Other; Grounds Maintenance Workers, All Other; Religious Workers, All Other; Artists and Related Workers, All Other; Textile, Apparel, and Furnishings Workers, All Other; Gambling Service Workers, All Other; Transportation Workers, All Other; Extraction Workers, All Other; Entertainment Attendants and Related Workers, All Other; Woodworkers, All Other; Underground Mining Machine Operators, All Other; Agricultural Workers, All Other; Logging Workers, All Other; Rail Transportation Workers, All Other; Communications Equipment Operators, All Other.

(h) **Home Health and Personal Care Aides:** This SOC Code is broken out into two occupations: Home Health Aides and Personal Care Aides. We added the tasks from both occupations.

(i) **Property Appraisers and Assessors:** This SOC Code is broken out into two occupations: Appraisers and Assessors of Real Estate and Appraisers of Personal and Business Property. We added the tasks from both occupations.

(j) **Miscellaneous Assemblers and Fabricators:** This SOC Code is broken out into two occupations: Assemblers and Fabricators, All Other and Team Assemblers. We match to Team Assemblers since Assemblers and Fabricators, All Other is a general occupation category without component tasks.

(k) **Electrical, Electronic, and Electromechanical Assemblers, Except Coil Winders, Tapers, and Finishers:** This SOC Code is broken out into two occupations: Electrical and Electronic Equipment Assemblers and Electromechanical Equipment Assemblers. We added the tasks from both occupations.

(l) **First-Line Supervisors of Transportation and Material Moving Workers, Except Aircraft Cargo Handling Supervisors:** This SOC Code is broken out into four occupations: First-Line Supervisors of Helpers, Laborers, and Material Movers, Hand, First-Line Supervisors of Material-Moving Machine and Vehicle Operators, First-Line Supervisors of Passenger Attendants, and First-Line Supervisors of Transportation Workers, All Other. We added the tasks from First-Line Supervisors of Helpers, Laborers, and Material Movers, Hand, First-Line Supervisors of Material-Moving Machine and Vehicle Operators, and First-Line Supervisors of Passenger Attendants. First-Line Supervisors of Transportation Workers, All Other is a general occupation category without component tasks.

2. **Missing Task Ratings.** There are 36 SOC-6 occupations which do not have any task rating in O*NET 28.3 or 29.0. These correspond to 34 SOC-4 occupations.[15] Among these, 2 SOC-4 occupations (Data Scientists and Web and Digital Interface Designers) have task ratings for some of the component SOC-6 occupations which allow us to compute the Adjusted and Weighted Task Share measures. For the rest of the 32 SOC-4 occupations that have no O*NET task ratings, we cannot compute the Adjusted or Weighted Task Share measure. Instead, we proxy the Weighted Task Share as follows: for each combination of 4-digit SOC occupation and task, we calculate the number of times the task appears (i.e., task frequency) for the occupation and divide by the sum of task frequency of all tasks of that occupation. For example, the 4-digit SOC occupation "Special Education Teachers, Kindergarten and Elementary School" combines two 6-digit SOC occupations (*Special Education Teachers, Elementary School* and *Special Education Teachers, Kindergarten*) has 43 unique tasks. Among these 17 tasks appear twice. Thus, the sum of task frequency across 43 tasks is 60. For each task that appears once, the proxy Weighted Task Share is $1/60 = 0.0017$, and for each task that appears twice, the proxy Weighted Task Share is $2/60 = 0.0033$.

### A.8.6 VALIDATING THE DIGITAL TASKS MEASURE

We benchmark our "knowledge work" classification method against the task-content framework of Acemoglu & Autor (2011).

---

[15]These SOC-4 occupations are: Aircraft Service Attendants, Bus Drivers, School, Calibration Technologists and Technicians, Cardiologists, Crematory Operators, Data Scientists, Disc Jockeys, Except Radio, Emergency Medical Technicians, Emergency Medicine Physicians, Entertainment and Recreation Managers, Except Gambling, Financial and Investment Analysts, Financial Risk Specialists, First-Line Supervisors of Entertainment and Recreation Workers, Except Gambling Services, First-Line Supervisors of Security Workers, Fundraising Managers, Health Information Technologists and Medical Registrars, Hydrologic Technicians, Legislators, Lighting Technicians, Medical Records Specialists, Orthopedic Surgeons, Except Pediatric, Paramedics, Pediatric Surgeons, Project Management Specialists, Public Relations Managers, Sales Representatives of Services, Except Advertising, Insurance, Financial Services, and Travel, School Bus Monitors, Shuttle Drivers and Chauffeurs, Software Developers, Special Education Teachers, Kindergarten and Elementary School, Substitute Teachers, Short-Term, Taxi Drivers, Teaching Assistants, Except Postsecondary, Web and Digital Interface Designers.

We further contextualize this validation against recent empirical work examining how AI systems are already being applied to economic tasks in practice. Tomlinson et al. (2025) measure the applicability of generative AI to occupations by mapping usage patterns onto O*NET task descriptions, while Handa (2025) analyze millions of real-world Claude interactions to understand which occupational tasks users delegate to AI systems. While these studies focus on observed AI usage rather than model capability, their findings are consistent with the patterns we observe when benchmarking our digital-task measure against the Acemoglu and Autor framework that AI use is concentrated in cognitively intensive information-processing tasks.

The framework in Acemoglu & Autor (2011) is based on the U.S. Department of Labor's O*NET survey, which collects data on the activities, work "content", and abilities required for each occupation. The framework aggregates these measures into five scores:

1. Non-routine cognitive: Analytical.

2. Non-routine cognitive: Interpersonal.

3. Routine cognitive.

4. Routine manual.

5. Non-routine manual physical.

Each score is computed as a composite measure of select O*NET "Importance" scales. For example, the "Non-routine cognitive: Analytical" score for each occupation is computed by summing the (normalized) values of the "Analyzing data/information" work activity, the "Thinking creatively" work activity, and the "Interpreting information for others" work activity. A high numerical value for an occupation for a given score indicates that the occupation relies heavily on that type of work.

We compute the Acemoglu & Autor (2011) scores for each occupation and then compare them with our measures of knowledge work (that is, the share of digital tasks and a binary "knowledge work" indicator for each occupation).

In our first set of results, we compare each Acemoglu & Autor (2011) task-content score with the share of digital tasks in an occupation. The patterns are clear: occupations with higher digital-task shares score systematically higher on the non-routine cognitive dimensions and lower on the manual dimensions. In other words, the more an occupation relies on digital tasks, the more it resembles cognitive, non-routine work.

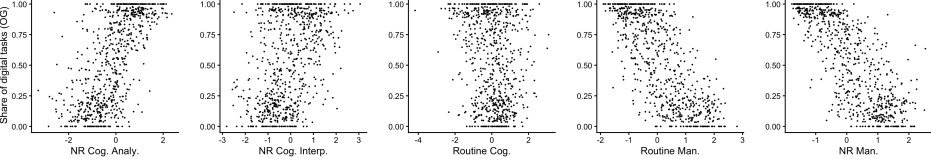

Figure 17: Distribution of occupations and task contents

In our second set of results, we look at the relationship between the Acemoglu & Autor (2011) scores and our binary measure of "knowledge work." In the following figure, we plot each occupation's value for each score, and color occupations by the paper's knowledge-work classification: blue for occupations identified as knowledge work and red for all others. The pattern is again clear– knowledge-work occupations cluster at the top of the non-routine cognitive distributions and at the bottom of the routine and manual distributions. Taken together, these results suggest that our digital-task classification is closely aligned with the economic literature on cognitive/manual work.

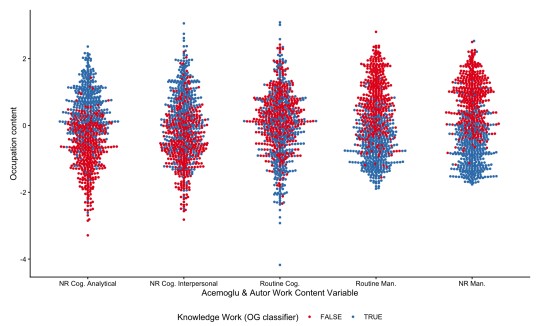

Figure 18: Scatterplot of digital tasks and task contents

