# OpenReview forum: "GDPval: Evaluating AI Model Performance on Real-World Economically Valuable Tasks"
_ICLR.cc/2026/Conference — ICLR 2026 Poster_

### Official Review · Reviewer_JEiu · 2025-10-19

**Soundness:** 2
**Presentation:** 1
**Contribution:** 4
**Rating:** 8
**Confidence:** 5

**Summary:**

The paper describes GDPval, a benchmark of 1320 occupational tasks written and reviewed by professionals in the relevant occupation. Each task involves the creation of a deliverable file (document, slide deck, video, etc). 7 LLMs were tested on GDPval and their outputs were scored by relevant professionals compared to professionally-created deliverables. The best-performing model achieved a win rate of about 45% against human experts. In supplementary analyses, the authors analyze the types of errors performed by different models, how reasoning effort contributes to performance, and how much improvement can be gained by prompt tuning. They also develop an automated grader that nearly achieves the same agreement on GDPval outputs with human graders as humans.

**Strengths:**

1. This is an impressive and ambitious research project.
2.The question is timely and of very high importance.
3. Gathering realistic tasks from professionals is a much better way of benchmarking economic potential of AI than existing approaches.
4. The amount of time, effort, and money needed to develop, review, and evaluate the tasks is very high.

**Weaknesses:**

1. The quality of the paper does not live up to the scale and ambition of the methods, with typesetting issues (see lines 075, 262, 288, 294), many missing details, and figures with no explanation or references.
2. The paper is sparse on details of how the models were run (how do these LLMs edit videos or produce CAD files?)
3. The prompts (both for GPDval tasks and the other classifiers run), model outputs, and human reviews do not appear to be available
4. The discussion of related work is very brief, only a few sentences; also, missing a few of the most important related papers:
	- Eloundou et al. "GPTs are GPTs: Labor market impact potential of LLMs." Science 384.6702 (2024): 1306-1308.
	- Tomlinson et al. "Working with AI: Measuring the applicability of generative AI to occupations." arXiv preprint arXiv:2507.07935 (2025).
	- Handa et al. "Which economic tasks are performed with AI? evidence from millions of Claude conversations." arXiv preprint arXiv:2503.04761 (2025).


### Overall evaluation
This research should obviously be published--it's very important and impactful, and the authors deserve credit and recognition for this work. However, the effort put into the paper falls short of the effort put into the research. The paper really should have some more time and care put into the writing: missing prompts and details should be added, all figures should be referenced and described, and related works should be fleshed out. More information should be provided as to the execution and outputs of the models. It's also not clear to me that ICLR is the best venue for this research. Despite that, I have to give the paper a high score simply due to its importance and impact. However, the authors should really improve the paper before any camera ready so it lives up to the impact of the research.

**Questions:**

1. The tuned prompt in Appendix A.3 implies the GPT-5 agent has significant computer-use ability, but the footnote on page 2 says other LLMs were interacted with through the UI; why was GPT-5 evaluated in a different context?
2. Relatedly, are all of the models listed in Figure 6 capable of outputting videos, slide decks, CAD files, PDFs etc?
3. Figure 3 is never referenced or described. What is the sandbox? Does an expert contribute multiple tasks, only the first of which goes through the iterative sandbox? Who are the "occupational experts" mentioned in the second stage?

### Comments
- The paper is missing prompts for the task-level "digital" classifier and the occupation-level "knowledge work" classifier; without carefully reading the appendix, the fact that there were two classifiers was confusing (the paper mentions both digital tasks and knowledge work, without making the distinction clear)
- O*NET isn't developed by the BLS, so calling them "U.S. Bureau of Labor Statistics Work Activities" in the abstract isn't accurate. (To be pedantic, it's developed by the North Carolina Department of Commerce under sponsorship from the Employment and Training Administration, part of US DoL)
- For percentage of digital tasks, the main text should reference the appendix section where the weighting method is described (page 21). (this weighting method is also a bit strange, since the 1-7 frequency codes have categorial meanings: 1="yearly or less", ..., 7="hourly or more")
- the note about fig 12 being on a different version of the test set should be in the figure caption, as this was confusing
- the more standard names for "SOC-4" and "SOC-6" codes are "SOC codes" and "O*NET-SOC codes," as one is not simply a truncation of the other. Was the BLS-provided crosswalk used to do the mapping between the two taxonomies? (https://www.bls.gov/emp/documentation/crosswalks.htm)

---

> ### Author Response · Authors · 2025-11-22
> **Comment 1/2**
>
> We thank the reviewer for the thoughtful and encouraging assessment. We greatly appreciate the recognition of the project’s ambition and impact. Below we address each point and in the camera ready we incorporate all requested clarifications and suggestions.
>
> 1. Unsightly formatting, missing details and references. Thank you, we completely agree that the formatting in our submitted paper was not up to standards! We have fixed all the formatting issues you identified (lines 075, 262, 288, 294), added missing figure references and captions, expanded the methodological details in the appendix, and fixed a number of other final paper issues in the updated version. We apologize for not doing this earlier!
> 2. Sparse details on how the models were run. We agree this section needs more clarity. Getting models to reliably read and output a wide variety of file types is challenging.For OpenAI models, we were able to sample files from OpenAI’s APIs. At the time of our submission, sampling files from Claude Opus 4.1, Gemini 2.5 Pro, and Grok 4 was not possible due to API file-handling issues (e.g., too-large file sizes, unsupported file types), so we sampled completions via their first party platforms (e.g., Claude Files UI) in order to solicit the best possible performance. But sampling from the UIs is more time consuming and lacks time and token-cost data similar to an API. Since our submission, all platforms have improved the API experience, and we are rerunning Opus, Sonnet, Gemini, and Grok via their APIs to add to the camera-ready, if accepted. In the camera-ready we also added additional detail on: 1) the open-source packages provided to the model to attempt file creation (appendix section A.7.5) 2) win rates by file type (A.2.4), 3) additional detail on failure modes by model which were often related to file handling (A.2.6), and 4) additional detail on multimodal capabilities, file handling, and UI vs. API differences across models (A.3). We also note that, because these results are on the open source set, they include fewer audio, video, and CAD files than the full set, because those file types were most difficult to scrub of potentially identifying information about the task writer.
> 3. The prompts, model outputs, and human reviewers do not appear to be available. We open-sourced a gold set of tasks and reference files on Hugging Face (under dataset: GDPval) and make our grader available so anyone can submit samples for grading. Though we did not open source model outputs, these can be obtained since the prompts and reference files are opensourced and we only sampled publicly available models. We added to the the paper the prompt that we used to classify O*NET tasks for each occupation as primarily digital/physical and the prompt that we used to elicit capabilities (A.3).
> 4. Missing related papers and brief related work section. Thank you for identifying key missing references. We added these papers to the Introduction to further contextualize and distinguish our approach to studying the potential labor impacts of frontier AI models. We reference additional related work in more detail in the Appendix, especially in A.8.6. “Validating the Digital Tasks Measure.”
> 5. Why was GPT-5 evaluated in a different context? See number 2.
> 6. Relatedly, are all of the models listed in Figure 6 capable of outputting videos, slide decks, CAD files, PDFs etc? See number 2.
> 7. Figure 3 is never referenced or described. What is the sandbox? Does an expert contribute multiple tasks, only the first of which goes through the iterative sandbox? Who are the "occupational experts" mentioned in the second stage? We added a pointer to the appendix from Figure 3 where we describe the review process in more detail, including the sandbox, which is a part of the onboarding process where experts submit their first task and receive iterative feedback from a trained reviewer. Once an expert had a signed off sandbox task, they submitted the rest of their tasks in the regular review pipeline. Experts contributed multiple tasks, and multiple experts submitted tasks for each occupation. “Occupational experts” were the experts from different occupations that we recruited to submit tasks. Some of the experts were promoted to occupational reviewers or generalist reviewers.

---

> ### Author Response · Authors · 2025-11-22
> **Comment 2/2**
>
> 8. The paper is missing prompts for the task-level "digital" classifier and the occupation-level "knowledge work" classifier. We added a table of all 44 occupations and their digital scores. We also added a table with a single occupation, its representative tasks, and classifier outputs to demonstrate the methodology for calculating the occupation digital score with the task level classifications. The full table of all occupations and all of their component tasks is ~20k rows and too long to add to the appendix, so we are including it as supplemental material. We also further clarified the distinction between the occupation-level digital score (which is computed) and task-level classifications and included the prompt we used to classify tasks.
> 9. O*NET isn't developed by the BLS, so calling them "U.S. Bureau of Labor Statistics Work Activities" in the abstract isn't accurate. (To be pedantic, it's developed by the North Carolina Department of Commerce under sponsorship from the Employment and Training Administration, part of US DoL). Thank you for catching this glaring error! We fixed this in the camera ready.
> 10. For percentage of digital tasks, the main text should reference the appendix section where the weighting method is described (page 21). (this weighting method is also a bit strange, since the 1-7 frequency codes have categorial meanings: 1="yearly or less", ..., 7="hourly or more"). Thank you for this note. We added a pointer to the appendix at the bottom of the main paper section. Regarding the frequency ratings, our intention was not to treat these values as literal counts, but rather as an ordinal proxy for relative task importance within an occupation. Even though the categories are discrete (“yearly or less,” … “hourly or more”), a rating of 7 unambiguously indicates that a task is performed more often and is typically more central to the day-to-day work of that occupation than a task rated 1. Using normalized frequency ratings as one component of our composite Adjusted Task Score therefore preserves the intended ordering of task relevance in O*NET. That said, we agree that the categorical structure is worth clarifying, and we will explicitly note this in the appendix.
> 11. The note about fig 12 being on a different version of the test set should be in the figure caption, as this was confusing. Thank you, we updated the plot to make this more clear.
> 12. "SOC-4" and "SOC-6" codes. We distinguish between SOC-6 and SOC-4 because these levels meaningfully differ in granularity. SOC-6 (O*NET-SOC) occupations have task-level ratings, whereas many SOC-4 (BLS SOC) aggregates do not have their own task lists and must be constructed by combining constituent SOC-6 occupations. We used the BLS-provided SOC-6 to SOC-4 crosswalk to perform all aggregations. This distinction is operationally important for how we compute task weights, and we will clarify this in the camera-ready.
>
> We thank the reviewer for the constructive feedback. We will substantially improve clarity, add missing prompts and details, strengthen discussion of model execution, and correct terminology and references.

---

> > ### Comment · Reviewer_JEiu · 2025-11-23
> >
> > Thanks for addressing the issues I raised! I continue to think this is an important contribution and advocate for acceptance, especially with the addition of experimental details like prompts and file creation capability for models.

---

### Official Review · Reviewer_ky9x · 2025-11-01

**Soundness:** 3
**Presentation:** 2
**Contribution:** 3
**Rating:** 4
**Confidence:** 2

**Summary:**

This paper introduces GDPval, a new benchmark to evaluate the performance of large AI models on real-world economically valuable tasks.

**Strengths:**

* This work introduces a benchmark (GDPval) that directly connects model performance to real-world economic value. The framing around professional tasks across multiple industries gives the research strong societal and economic relevance, and it has clear potential to become a foundational evaluation backbone for assessing AI models’ productivity capabilities.

* The benchmark covers 9 major sectors and 44 occupations, representing a large portion of GDP-contributing work activities. The inclusion of tasks curated from professionals with extensive industry experience makes the evaluation dataset both realistic and diverse.

* The paper reports quantitative progress of recent LLMs on GDPval but also analyzes key performance drivers such as reasoning effort, context size, and scaffolding, offering actionable insights into how models achieve expert-level deliverables.

**Weaknesses:**

* While the authors claim that they will open source their benchmark, I do not see any anonymous link or supplementary that contains the related benchmark file. Given that this benchmark is the major contribution of this paper, the authors should make sure the benchmark will be open-source.

* Since the benchmark focuses mainly on U.S. GDP-related activities and professional writing tasks, it may underrepresent other economically relevant domains (e.g., manufacturing, logistics, physical or multimodal tasks). Broader coverage would improve generalizability.

* The grading pipeline and automatic scoring system are promising but may introduce bias or subjectivity, particularly when comparing AI outputs to human experts. More discussion on inter-rater consistency and rubric validation would strengthen the reliability of the results.

**Questions:**

* The paper mentions “We fine-tune GPT-5 on GDPval data and measure clear improvements in human win-rate” in the conclusion part. Given that GPT is a close-looped LLM, can the authors provide further explanation about the meaning of fine-tuning here?

---

> ### Author Response · Authors · 2025-11-22
> **Comment 1/2**
>
> We thank the reviewer for the thoughtful and encouraging feedback and appreciate the recognition of the benchmark’s importance.
>
> 1. While the authors claim that they will open source their benchmark, I do not see any anonymous link or supplementary that contains the related benchmark file. Given that this benchmark is the major contribution of this paper, the authors should make sure the benchmark will be open-source.
>
> We agree that open-sourcing the benchmark is essential for reproducibility, and should be a pre-requisite to our acceptance. We have open-sourced the gold set of tasks and reference files on Hugging Face (under datasets: GDPval), where they hit #1 daily downloaded on Hugging Face in the days after publication, and also provide a free grader service so external parties can grade their samples using the same autograder as discussed in the paper. The dataset has already been downloaded over 10K times, and the many submissions to our autograder service have already been processed. The full URL of both links (dataset, autograder) would identify us, as the domain names give away our institution, so we chose to not link them in our submission. We added a .parquet file of the tasks to the .zip in Supplementary Material.
>
> We hope that the open-sourcing of our benchmark will help revise our score for acceptance.
>
> Note: we maintain a holdout set that we keep private to avoid contamination, so the full set of tasks was not open-sourced.
>
> 2. Since the benchmark focuses mainly on U.S. GDP-related activities and professional writing tasks, it may underrepresent other economically relevant domains (e.g., manufacturing, logistics, physical or multimodal tasks). Broader coverage would improve generalizability.
>
> We focused the initial version of GDPval on tasks that could feasibly be completed by a knowledge worker with access to the files provided, widely available tools, and the internet. As discussed in the limitations section, the current benchmark focuses on digital knowledge-work tasks done on a computer and therefore does not cover many economically valuable *physical* or *manual labor* tasks. As suggested by anther reviewer, we also revised our abstract to state that we cover “real-world, economically valuable digital knowledge-work tasks” to make this focus on digital knowledge work explicit. However, as mentioned on line 079, we *do* include substantial coverage of *multi-modal* tasks (e.g., CAD design files, photos, video, audio, social media posts, diagrams, slide decks, spreadsheets, and customer support conversations). We wholeheartedly agree that broader coverage (including for physical tasks) is an important direction for future work. Still, many of the most-GDP relevant occupations (occupations that represent the largest share of total employment and dollars earned) are knowledge work occupations, so we believe a realistic digital knowledge work benchmark is an important contribution.

---

> ### Author Response · Authors · 2025-11-23
> **Comment 2/2**
>
> 3. The grading pipeline and automatic scoring system are promising but may introduce bias or subjectivity, particularly when comparing AI outputs to human experts. More discussion on inter-rater consistency and rubric validation would strengthen the reliability of the results.
>
> We included additional detail on the autograder in appendix A.6, but agree that additional detail would strengthen this paper. We agree that transparency around rubric construction and inter-rater consistency is necessary for interpreting our results, especially given the subjectivity of many GDPval tasks. In response, we added explicit pointers in the main text to our grading validation analyses, and expanded our appendix with additional methodological detail. We hope these changes substantially improve our paper, if accepted.
>
> Regarding rubric validation (as we now detail in the paper): Rubrics are used by the autograder only (not by human expert graders). For the validation pipeline, these rubrics used by the autograder are first written by experts trained in rubric writing, then reviewed by another human expert from the same occupation as the original task writer to improve quality. We iteratively improved these rubrics using a model prompted for rubric refinement, and every revision was checked again by an expert rubric writer. This process helped ensure that the rubrics were clear, consistent, and aligned with domain expectations. However, many GDPval tasks are highly subjective or have many valid completions. Therefore, one of the limitations of the rubrics is that they need to allow fair grading of a wide range of potential solutions, and this is difficult for traditional rubric writers who are trained to look for objective criteria in the prompt, not to search the entire solution space for all objective components of an ideal solution. We used models to help search the solution space, but models are less capable than the human rubric writers at writing objective criteria and checking them, and at keeping rubric items self contained. Therefore, we think there is more work to be done here to ensure the rubrics truly capture subjective criteria. We added information about the rubric writing process in the camera-ready (Appendix Section A.7.3).
>
> For more discussion of the human and autograder inter-rater agreement, see Appendix section A.7.1. The final autograder model achieved 66% agreement with human expert graders, which was 5% below human expert inter-rater agreement of 71%, which is why we believe results from the autograder are significantly limited compared to those of the experts. We also include more detail on different metrics to calculate inter-rater agreement, and how to handle ties, in our paper.
>
> We acknowledge that both the autograder and the human-trainers can exhibit bias in their grading. For humans, it’s often easy to tell that an output is model generated because of artifacts like em-dashes or glaring visual errors that a human would not make (like overlapping text and images in a powerpoint deck). For models, they tend to prefer their own outputs to the outputs of other models.
>
> Overall, we still think that using expert human graders is the current best "gold grading standard" here, because in real life, a human manager would be grading the quality of the work of their report/employee. Ultimately, grading of these types of tasks is subjective, and that is why it is so interesting to see how models perform when graded by true experts in the field who would be judging their own employee's contributions. Therefore, we think the results from the expert human grading are interesting and contribute to an improved understanding of model capabilities.
>
> 4. The paper mentions “We fine-tune GPT-5 on GDPval data and measure clear improvements in human win-rate” in the conclusion part. Given that GPT is a close-looped LLM, can the authors provide further explanation about the meaning of fine-tuning here?
>
> Thank you for catching this ambiguity. We removed the reference to “fine-tuning” from the conclusion. Our experiments do not involve weight updates to GPT-5; the improvements we report come from prompting and scaffolded inference described in Section 3.3. We updated the text accordingly to avoid any implication of closed-model fine-tuning.
>
> We thank the reviewer again for engaging deeply with the work. We believe your suggestions substantially improved the final paper and hope to receive a revised score.

---

### Official Review · Reviewer_uLMP · 2025-11-02

**Soundness:** 2
**Presentation:** 2
**Contribution:** 2
**Rating:** 2
**Confidence:** 4

**Summary:**

The authors introduce an evaluation dataset for "real-world economically valuable tasks" (from occupations with predominantly digital tasks) and show how current frontier models perform on these tasks compared to experts.

**Strengths:**

- The authors aim to provide a more granular evaluation to capture the economic impact of AI, an area that is currently lacking data
- The experts that the authors recruited had extensive experience and were very carefully vetted
- The authors provide a smaller open-source evaluation dataset as well as an initial version of an automated grader for these tasks
- The writing is overall clear.
- The iterative task review process with multiple expert review rounds (and an initial sandbox round with additional feedback) was very well thought-through and designed.

**Weaknesses:**

Note: I am willing to update my score but currently, there is a lot of methodological information missing in the paper that prevents me from fully assessing the work. Happy to revise my score once more details are provided.

- Line 091: Where do these time estimates come from?
- Line 106: How was the 5% threshold chosen?
- Given that in the current version only jobs with predominantly digital tasks were included, the premise of GDPval seems overstated. The abstract states that GDPVal is “an evaluation assessing AI model capabilities on real-world economically valuable tasks” but doesn't specify that it is limited to occupations with predominantly digital tasks only.
- The paper only states the set of 44 occupations but not the tasks/task descriptions for each of these. I strongly encourage the authors to add, to the appendix, a list of high-level summaries of the tasks and to which occupation they belong to allow for a more meaningful assessment of the paper. For example, I find it puzzling that "registered nurse" is in the 44 occupations considered and would have needed a summary of tasks added for this occupation to verify whether this classification made sense. I understand the authors’ concerns that they do not want to make all task specifics/prompts/files/etc. available due to potentially identifying information of expert, but they can make a high-level description of the task and the occupation it belongs to available.
- Figure 7(a) and (b): Why were OpenAI models singled out here? How does this analysis look like for the Claude or Gemini families? Overall, most analyses in the paper single out OpenAI models and don’t allow for an accurate understanding of frontier model capabilities. Given that the full tasks aren’t open-sourced and the analysis hence cannot be completed for other models by third parties, I kindly ask the authors to provide the more detailed analysis for non-OpenAI models, too, to allow for better understanding and comparability of frontier model performance. E.g., how would Figure 7(b) look like for Claude models (which was overall the best-performing model according to Fig. 6? Similarly, in Section 3.3, this analysis was only done for improving formatting. How much of a performance increase would models like Claude or Gemini have gotten if the model was prompted to strictly follow instructions?
- The reported details for the human baseliners and recruited experts are insufficient. For example, there is largely no information about the recruitment process given. Also, the paper only states that experts were “well compensated” but given the impact of compensation of performance, this should be made transparent. See this paper for the level of detail required for human baseline reporting: https://arxiv.org/pdf/2506.13776. Relatedly, how were occupational experts (line 206) sourced?
- It’s unclear to me what “their requests” mean in line 172. Who assigned them “their requests”? Were multiple experts from the same occupation tasked with creating tasks? If this wasn't the case, it’s unclear to me how content validity (i.e., coverage) was ensured for each occupation.
- It feels like the claim in line 287/288 that GPT-5 excelled in particular on accuracy is overstated. From Fig 8, the % of total examples of failure modes seems too close for accuracy to say that a model “particularly excelled” (within +/- 1% for Claude and GPT-5) and in addition, Grok 4 showed the same % as GPT-5 so I don’t understand why GPT-5 is being singled out here. Are these significant differences?
- The human uplift study design has not been fully explained. How were scenarios picked? Why were only OpenAI models tested if Claude Opus 4.1 was better overall?
- It’s unclear if the authors had an IRB or applied for an IRB exemption, which would be necessary for the experimental setup with significant human involvement. Please clarify.
- Can you provide quantitative data for the clustering pipeline? I.e., a table where there is quantitative information per model on how often the rationale fell into one of the buckets listed in line 311 ff., how often the rationale was unclear, etc. Otherwise it’s hard to judge the true differences across models. It’s also unclear if these differences were statistically significant, please add this information.
- Line 335: Note that there is a bias if you use the same model to judge the same model (e.g., GPT-5 judge to judge GPT-5 output); it’s best practice to at least compare the judgement to other judge models, see this paper: https://arxiv.org/abs/2404.13076. I'd suggest to at least add this to the limitation section.

Nit picks (only affected presentation score):
- The authors talk at multiple points about the "gold set" but it's unclear how the tasks for the gold set were selected.
- There seems to be a reference error in line 075.
- Lines 155ff.: It’s unclear to me how the list of prior employers is relevant for describing the experts.
- The formatting in lines 216/217 seems off, shouldn’t this be a full line?
- The format looks again off in lines 260/261, these should be full lines. Did you change anything about the underlying template?
- Figure 8 requires a lot of zooming in to be readable.
- Line 327/328: this statement is too qualitative. How much did performance improve? I'd suggest to least add a reference to the corresponding figure here.
- The paper lacks a proper related work section. While the authors very briefly mention previous work in one line in the introduction, they don’t sufficiently explain how their work is novel/different.

**Questions:**

Questions:
- Can you explain the notion of “significantly limited” in 213?
- I don’t understand how the last sentence in line 290 squares with the rest of the paragraph. The previous sentence seems to say that only 47.6% of model deliverables were as good or better than the human deliverable so I don’t understand where the “just over half the tasks” statement comes from. Could you help me understand that, please?
- Why couldn’t cost estimates for Claude, Gemini, and Grok be obtained? API cost for estimates should be available for all three models, no?

**Details Of Ethics Concerns:**

It’s unclear if the authors had an IRB or applied for an IRB exemption, which would be necessary for the experimental setup with significant human involvement in my opinion.

---

> ### Author Response · Authors · 2025-11-22
> **Comment 1/5**
>
> We sincerely thank the reviewer for the constructive suggestions and address each one below.
>
> 1. Line 091 Where do these time estimates come from?
>
> Thank you for the question. This was buried in the paper previously. We previously mentioned on page 12 and footnote 5 in Appendix ("Speed and Cost Analysis, Continued") from our original submission that "Human expert professional completion time $H_T$ is the time taken by a human expert professional to complete a task, based on validated self-reported time to complete" and "During submission, experts self-reported the real-world time required to complete each task. Multiple occupational reviewers independently validated these times, correcting errors. Because times were self-reported, it is possible that experts under-estimated or over-estimated time taken." To further elaborate, time estimates (estimated time taken to complete the task in real life) are self-reported by occupational experts who created the tasks. Each time estimate then went through at least 3 rounds of multi-stage review, including a review by an expert in the same occupation. These reviews flagged potential errors (e.g., when numbers seemed like a typo, or if an estimate seemed unrealistically high or low given another expert's experience with time needed for the task), and task authors revised accordingly. We updated the paper to clarify this methodological detail in the main text, and we thank the reviewer for suggesting the addition.
>
> Because our tasks were drawn from real-world work (already completed work tasks, completed at experts’s actual jobs in a real-world setting) across over 40 occupations, there was no other option to estimate time taken besides self-reporting. Task authors were encouraged to try to be as accurate as possible, and (if available) could use time-stamps from their real lives to help themselves put together the estimate, but self-reporting is an inherent methodological limitation (as mentioned in Limitations). If we had asked the experts to complete tasks fresh and timed them ourselves, we could have guaranteed that the estimates were perfect; however, this would have made the tasks unrealistic (i.e., completed in an artificial setting for this study) and also would have been too expensive given some of the best long-horizon tasks took days or weeks to complete. Therefore, we believe our approach was the best way to ensure real-world task realism (which was the primary objective of our benchmark) while maintaining quality control on self-reported times.
>
> 2. Line 106 How was the 5% threshold chosen?
>
> We started with industries constituting greater than 5% of GDP to prioritize representation of the largest industries while respecting constraints of our data budget. The choice was not meant to imply that lower-contribution industries are unimportant but rather was a practical choice due to our budget. In future GDPval iterations, we’re adding sectors that didn’t make the cut the first time (including those that constitute <5% of GDP). This is clarification is now in the paper.
>
> 3. Only predominantly digital tasks were included.
>
> We agree the abstract should reflect the digital nature of the work and revised it to “real-world, economically valuable digital knowledge-work tasks” to make this limitation explicit.
>
> 4. The full set of occupations, their component scores, and digital task classifications are not in the paper. I strongly encourage the authors to add, to the appendix, a list of high-level summaries of the tasks and to which occupation they belong to allow for a more meaningful assessment of the paper.
>
> We agree that more detail about the task level classifications and occupation digital scores would strengthen the paper. We added a table of all 44 occupations and their digital scores to the paper. We also added a table with a single occupation, its representative tasks, and classifier outputs to demonstrate the methodology for calculating the occupation digital score with the task level classifications. The full table of all occupations and their component tasks is ~20k rows and too long to add to the appendix, so we include it as supplemental material. We also further clarified the distinction between the occupation-level digital score (which is computed) and task-level classifications and included the prompt we used to classify tasks. We hope this makes clearer how the initial set of occupations were selected. In general, our methodology for the initial version of GDPval was to start with the highest-contribution sectors to GDP, and then prioritize the highest contributing knowledge work occupations within each industry. The initial cut of occupations in GDPval-v0 helped us prove out the concept, optimize our human data pipeline, and showcase early findings, but is not meant to imply that the initial selected occupations are the only important knowledge work occupations. We are adding more industries, occupations, and tasks in the next GDPval iteration.

---

> ### Author Response · Authors · 2025-11-22
> **Comment 2/5**
>
> 5. Figure 7(a) and (b) / Why were OpenAI models singled out?
>
> Getting models to reliably read and output a wide variety of file types is challenging. For OpenAI models, we were able to sample completions for all GDPval tasks from OpenAI’s APIs. At the time of our submission, sampling files from Claude Opus 4.1, Gemini 2.5 Pro, and Grok 4 for GDPval was not possible due to API file-handling issues (e.g., too-large file sizes, unsupported file types), so we sampled completions via their first party platforms (e.g., Claude Files UI) in order to solicit the best possible performance. But sampling from the UIs is more time consuming and lacks time and token-cost data similar to an API, so while an apples-to-apples comparison on performance was possible, a comparison on time and cost was not. Since our submission, all platforms have improved the API experience, and we are rerunning Claude Opus, Gemini, and Grok via their APIs to add time and cost analyses for these models to camera-ready, if accepted.
>
> 6. The reported details for the human baseliners and recruited experts are insufficient. For example, there is largely no information about the recruitment process given. Also, the paper only states that experts were “well compensated” but given the impact of compensation of performance, this should be made transparent. See this paper for the level of detail required for human baseline reporting: https://arxiv.org/pdf/2506.13776. Relatedly, how were occupational experts (line 206) sourced?
>
> We appreciate this request for more detail on the human baseliners who participated and were unaware of the linked human baseliner checklist. In the camera-ready, we added more information on recruitment, eligibility criteria, acceptance rates, compensation, training, and additional information from the human baseliners checklist, which we now cite in the paper as well (Handa et al., 2025). We include some of this new detail below:
>
> We recruited expert industry professionals to create realistic tasks based on their professional work experience. Experts were required to have a minimum of 4 years of professional experience in their occupation and a strong resume with a demonstrated history of professional recognition, promotion, and management responsibilities. The average expert had 14 years of experience in their occupation. We further required experts to pass a video interview, a background check, a training and a quiz to participate in the project. Experts were well compensated for their time and experience.
>
> Human baseliners and occupational experts were recruited for each occupation through postings on platforms like LinkedIn  and targeted outreach. Applicants were screened using resume review, an interview, and an occupation-specific questionnaire. Eligibility required at least 4 years of full-time experience in the target occupation and fluency in English (all tasks were written and graded in English). Approximately 10\% of applicants met these criteria and were selected; ~90\% were excluded based on insufficient experience. Experts were predominantly based in the United States, United Kingdom, and Canada. All human contributors were paid hourly rates competitive for their occupation and geography. Average compensation levels were >$120/hr, and varied by occupation to be competitive with market rates. Compensation varied by occupation, prior experience, and region, rather than being uniform across contributors. All experts who created tasks participated in live training calls covering grading standards, exemplar tasks, common errors, and expectations around deliverable quality. Contributors also had access to office hours and dedicated Slack channels for ongoing clarification and support throughout task writing and grading. Writers and graders were compensated for training time. No author served as a baseliner or grader.
>
> Experts determined the number of tasks they completed per session (instrument length was not fixed). Items were not randomized. Quality control was performed throughout data collection. Our staff monitored Average Handling Time (AHT), and reviewers flagged low-effort or low-quality submissions. Tasks failing quality thresholds were excluded from the gold set. More detail on review pipelines is in the review sections of the main text and Appendix. Human reviewers conducted multiple rounds of review on each task.
>
> Humans and models did not use equivalent UIs: humans worked through a graphical interface, whereas models interacted through APIs or first-party UIs depending on file-handling reliability.
>
> We thank the reviewer for this suggestion and hope the increased detail is helpful.

---

> ### Author Response · Authors · 2025-11-23
> **Comment 3/5**
>
> 7. It's unclear to me what “their requests” mean in line 172. Who assigned them “their requests”? Were multiple experts from the same occupation tasked with creating tasks? If this wasn't the case, it’s unclear to me how content validity (i.e., coverage) was ensured for each occupation.
>
> Tasks (which include a deliverable and a request) are based on real work performed by experts, which in the paper we called “deliverables” (to refer to the final output) and “requests” (to refer to the prompt). In real life, the experts would have been assigned requests by their places of work. All tasks included in GDPval draw from real-world work.
>
> Yes, multiple experts contributed tasks per occupation and then mapped their tasks to the representative O*NET tasks. Though we do not claim full coverage of all tasks in each occupation, the resulting submitted tasks did cover majority of task types.
>
> 8. It feels like the claim in line 287/288 that GPT-5 excelled in particular on accuracy is overstated. From Fig 8, the % of total examples of failure modes seems too close for accuracy to say that a model “particularly excelled” (within +/- 1% for Claude and GPT-5) and in addition, Grok 4 showed the same % as GPT-5 so I don’t understand why GPT-5 is being singled out here. Are these significant differences?
>
> Thank you for this note. We included this line originally out of completeness (we were noted which model was the best in each category, noting also Claude where it was the best in formatting, for example) -- we were not singling any models out arbitrarily. But we take the reviewer's point that all of these are too small of a difference to be worth calling out, and edit in the camera-ready version to only include the plot. Moreover, as further detailed in the failure modes section in later parts of the comments from us, we are now breaking out the claims into specific failure categories rather than generalizing at such a high level. We also add a new analysis that shows that GPT-5 was the best performing model on text output format while Claude Opus 4.1 was best performing on all other file formats, further demonstrating how Claude Opus 4.1 was able to perform best overall on GDPval due to superior performance on non-text file types (e.g., PPT, XSLX) and multi-modal formatting, which are quite important for knowledge work tasks.
>
> 9. The human uplift study design has not been fully explained. How were scenarios picked? Why were only OpenAI models tested if Claude Opus 4.1 was better overall?
>
> This design was further explained in detail on page 12-14 of our original submission in the "Speed and Cost Analysis, Continued" section of the Appendix, but we acknowledge that we should have made it clearer in the main body of the paper, which we have now updated. This was not a human uplift study, but rather a modeling exercise to study whether incorporating a model in the loop could have been (in a simulated setting) more time and cost effective compared to not using a model, under certain task completion assumptions (as detailed in the paper). The goal of this section was to study human vs human+model, not to compare cost and time across model families. We therefore used a single, representative model family (GPT-5), which was also the most accessible for repeated runs via the API. At the time, reliable cost and time data for Claude Opus 4.1, Gemini 2.5 Pro, and Grok 4 could not be collected due to API file-handling issues, and first-party UIs from these providers did not provide time or token cost data. We updated the section to state our design choices more explicitly, including with more detail in the main body of the paper, and are also re-running the gold set on the Claude/Gemini/Grok APIs to get this data (since these APIs now process files better), which we intend to include in the camera ready, if accepted. We believe this section does add value because traditional model benchmarks tend to focus on comparing performance of models versus human experts on overall performance, but not on speed and cost. We were excited to share some of the math behind how time and cost scenarios could be analyzed, and believe they add value to our paper by showcasing the overall potential for real-world knowledge work impact from these models.

---

> ### Author Response · Authors · 2025-11-23
> **Comment 4/5**
>
> 10. It’s unclear if the authors had an IRB or applied for an IRB exemption.
>
> Our study did not meet the definition of human subjects research because we were studying models, not human subjects. The deliverables in GDPval were already created in the course of normal work, and not collected specifically for this study. Experts served as professional contractors submitting and evaluating work outputs, analogous to standard data-labeling pipelines. We updated our ethics explanation in the camera-ready to clarify that we did not meet the definition of human subjects research.
>
> 11. Can you provide quantitative data for the clustering pipeline? I.e., a table where there is quantitative information per model on how often the rationale fell into one of the buckets listed in line 311 ff., how often the rationale was unclear, etc.
>
> The quantitative clustering results were plotted in Figure 9 in our original paper, but due to poor formatting on our part, we see that the plot was cut off. We understand why further detail on the exact numbers would be useful to asses this paper. As per this suggestion, we added additional per-model cluster data to more clearly quantify the failure modes in Appendix section A.2.6 and also made the plot more legible for the camera-ready version, if accepted. All expert human grades were required to have a rationale, so we are not missing rationales on any grades. All failure modes were bucketed in one of the following primary failure modes (since all reasons were related to accuracy, instruction following, or multi-modal formatting in some way). Summary below:
> | Failure mode                 | Description                                                                                                                                        | Claude Opus 4.1 | GPT-5-high | Gemini 2.5 Pro | Grok 4 |
> | ---------------------------- | -------------------------------------------------------------------------------------------------------------------------------------------------- | --------------: | ---------: | -------------: | -----: |
> | Produces factual errors      | Numbers, formulas, or factual statements in the deliverable are wrong or inconsistent with source data, undermining trust in the output.           |           2.68% |      2.22% |          3.43% |  3.18% |
> | Omits requested deliverables | Did not provide the requested deliverable file(s).                                                                                                 |           1.77% |      1.36% |          7.78% |  6.01% |
> | Omits required content       | The delivered file exists but leaves out major sections, data, or elements.                                                                        |           3.79% |      5.40% |          8.23% |  8.23% |
> | Omits required visuals       | Required images, charts, or media assets were not included, leaving the deliverable text-only and non-compliant with visual specifications.        |           2.42% |      2.47% |          3.23% |  3.43% |
> | Delivers wrong format        | A file was delivered, but it was formatted differently than the prompt specified.                                                                  |           0.56% |      0.81% |          4.19% |  4.90% |
> | Provides unusable files      | The delivered file is present but core functionality (formulas, dropdowns, links) is broken, so users cannot use it without repair.                |           0.96% |      0.81% |          1.36% |  0.86% |
> | Broken formatting            | The file is present but serious layout or formatting problems (text off-page, overlap, corrupt fonts) make the content hard or impossible to read. |           2.58% |      5.05% |          4.65% |  8.18% |
>
> 12. Line 335: Note that there is a bias if you use the same model to judge the same model (e.g., GPT-5 judge to judge GPT-5 output); it’s best practice to at least compare the judgement to other judge models, see this paper: https://arxiv.org/abs/2404.13076. I'd suggest to at least add this to the limitation section.
>
> We already discuss this limitation in the appendix (line 1057), citing Panickssery et al, but this is a good point to call out further, so we also added a pointer in the main text and our limitations section. We treat the human expert grader comparisons as the source of truth, as discussed in the main text of the paper. This further minimizes bias when comparing between models and more accurately represents a setting where a human manager is selecting the best deliverables.

---

> ### Author Response · Authors · 2025-11-24
> **Comment 5/5**
>
> 13. The authors talk at multiple points about the "gold set" but it's unclear how the tasks for the gold set were selected.
>
> The gold set is a subset of 5 tasks per occupation (220 tasks total) that we chose to be representative set by selecting, for each occupation, tasks that covered a broad range of O*NET tasks, were rated as high quality by reviewers, and were possible to scrub of any information that could potentially be used to re-identify the original task writer. We include this detail in the paper.
>
> 14. There seems to be a reference error in line 075.
>
> Thank you for the catch; the reference error in line 075 has been fixed.
>
> 15. Lines 155ff.: It’s unclear to me how the list of prior employers is relevant for describing the experts.
>
> We included prior employers to provide context on the professional caliber and industry experience of the experts who participated in task writing and grading. Because GDPval relies on high-skill work, we hope that showing that contributors came from well-known, high-performing firms helps establish the credibility and relevance of the experts’ backgrounds.
>
> 16. The formatting in lines 216/217 seems off, shouldn’t this be a full line? The format looks again off in lines 260/261, these should be full lines. Did you change anything about the underlying template?
>
> Thank you for catching this! Formatting in lines 216/217 has been fixed and in 260/261. Thank you for catching these issues; we completely agree that the formatting in our submitted paper was not up to standards and have improved it substantially.
>
> 17. Figure 8 requires zooming to be readable.
>
> We replotted with greater readability, which we are including in the camera-ready, if accepted.
>
> 18. The paper lacks a proper related work section. While the authors very briefly mention previous work in one line in the introduction, they don’t sufficiently explain how their work is novel/different.
>
> Thank you for this suggestion. We updated the introduction with additional references (Eloundou et al, Tomlinson et al, and Handa et al.) and language to explain the novelty of our benchmark from prior work that seeks to study the labor impacts of AI. We also already include on pages 1-2 aspects of this benchmark that differentiate it from prior benchmarks (namely: increased realism from using actual work tasks, more representative breadth by covering the majority of Work Activities tracked by O*NET for each occupation, multimodality across a variety of formats including CAD design files, photos, video, audio, social media posts, diagrams, slide decks, spreadsheets, and customer support conversations, and very long horizon work with the average task taking a full day to complete).
>
> 19. Can you explain “significantly limited” in 213?
>
> Yes, the autograder significantly underperforms experts. As mentioned in the main body of the paper (line 217), further detail is in appendix A.6.. Here, we detail the lower inter-rater agreement from the autograder compared to human experts, as well as other limitations of the grader (difficulties handling certain font packages, audio files, non-Python coding languages, and live internet access).
>
> 20. I don’t understand where the “just over half the tasks” statement comes from.
>
> Thank you for catching this typo! We updated to "just under half."
>
> 21. Why couldn’t cost estimates for Claude, Gemini, and Grok be obtained?
>
> To maximally elicit performance, we used the UI interface for these models, which at the time of submission was better at handling files than then API. However, in camera-ready (if accepted), we are also including a version using API completions, which also allows us to measure cost and speed for these models.
>
> We sincerely thank the reviewer for their feedback and hope our substantial revisions can help improve our score for acceptance to ICLR.

---

> > ### Comment · Reviewer_uLMP · 2025-11-28
> >
> > Dear authors, thank you for the detailed responses to my questions and concerns. I'm generally open to raising my score but would like to see the actual edits, could you upload the updated PDF with the changes outlined above? Thank you in advance!

---

> ### Author Response · Authors · 2025-12-02
> **Reply**
>
> Yes, updated the PDF + supplementary materials! We also added detail directly in the comments above too!

---

### Author Response · Authors · 2025-11-22
**Global Response**

Dear all,

We thank the reviewers for taking the time to review our work, and for all the constructive suggestions. We believe this helped significantly strengthen the paper and really appreciate the time and feedback, which we address in more detail below. We also appreciate the endorsements of our work, particularly the reviews stating that GDPval is "impressive," "ambitious," "timely," and has the "potential to become a foundational evaluation backbone" for economic impact of AI. Since putting our evaluation on Hugging Face (dataset: GDPval), we hit #1 trending on Hugging Face, with over 10K downloads in the first month. We hope this demonstrates the impact GDPval is already having on the research community.

We recognize also that our original paper submission had significant aesthetic and presentation issues and we deeply apologize for this. We have since fixed issues with formatting, added missing figure references and captions, and expanded on missing methodological details of the paper. We apologize for not doing this earlier!

In terms of content, we substantially increased methodological detail in the body of the paper (as opposed to keeping most of it in the Appendix), as requested by all reviewers. We hope this makes our work much more legible. Also, two of the plots (Figures 9a and 15) have been updated with additional data collected since the initial submission of the paper. Finally, we are also re-running GDPval on the API versions of Claude, Gemini, and Grok for updated time and cost analysis, now that the APIs can handle the file sizes and types in GDPval. We will update our paper with these results for the camera-ready, if accepted.

Thank you so much again to the reviewers for taking the time to review our work. We believe this helped make our paper much stronger and really appreciate the feedback.

Best,
Authors

---

### Author Response · Authors · 2025-12-02
**Comment for Area Chairs**

Dear Area Chairs,

Thank you so much for your help reviewing our submission. We are grateful to all reviewers for their detailed and constructive feedback, which substantially improved the paper.

Because the OpenReview issue prevented updates to scores, we provide the following brief summary for your consideration. Notably, Reviewer uLMP—the reviewer who gave us the lowest score (2)—explicitly stated that they would revise their score once details are provided: “I am willing to update my score but currently, there is a lot of methodological information missing in the paper that prevents me from fully assessing the work. Happy to revise my score once more details are provided," and "Dear authors, thank you for the detailed responses to my questions and concerns. I'm generally open to raising my score but would like to see the actual edits, could you upload the updated PDF with the changes outlined above? Thank you in advance!" We addressed all of their methodological concerns in the 5 rebuttal comments and revised submission, and request that you take the reviewer's stated intention into account. We go into more detail in rebuttal comments below on all the places we fleshed out the methodological information the reviewer requested.

Similarly, we addressed all 3 weaknesses from reviewer ky9x, who gave us a 4 for "marginally below the acceptance threshold. But would not mind if paper is accepted." The first weakness stated: "While the authors claim that they will open source their benchmark, I do not see any anonymous link or supplementary that contains the related benchmark file. Given that this benchmark is the major contribution of this paper, the authors should make sure the benchmark will be open-source." We clarified that we open-sourced the eval on Hugging Face under datasets: GDPval (where it reached #1 in weekly trending and hit over 10K downloads), and also added a zipped copy to the supplementary materials to avoid sharing a de-anonymized link, addressing the reviewer's concern about open-sourcing of tasks. Second, we clarified that we had tasks that are indeed multimodal, in response to the weakness about not having multi-modal tasks. Third and finally, the reviewer said, "More discussion on inter-rater consistency and rubric validation would strengthen the reliability of the results," which was a great point and that we added to the paper address the concern.

The final reviewer who gave us "8: accept, good paper (poster)" told us "Thanks for addressing the issues I raised! I continue to think this is an important contribution and advocate for acceptance, especially with the addition of experimental details like prompts and file creation capability for models."

We encourage the area chairs to read our detailed rebuttals below addressing these points. We believe this process helped make our paper much stronger and addressed the concerns of the reviewers.

Thank you!

Best,
Authors

---

### Meta-Review · Area_Chair_cyCN · 2026-01-02

**Summary:**

The paper introduces GDPval, a benchmark for evaluating large AI models on economically valuable, real-world tasks. To address the reviewers' concerns, the authors open-sourced the benchmark, clarified the inclusion of multimodal tasks, and added discussion on inter-rater consistency and rubric validation.

**Reviewer Concerns:**

Reviewers uLMP and JEiu have indicated in their post-rebuttal responses that their concerns have been addressed. Reviewer ky9x did not post a post-rebuttal response; however, after reading the authors’ rebuttal, I believe their major concerns have been addressed as well.

**Reviewer Scores:**

Reviewers uLMP and ky9x may increase their scores accordingly if they read the authors’ response and participate in the discussion.

---

### Decision · Program_Chairs · 2026-01-26

Accept (Poster)